# TA-GATES: An Encoding Scheme for Neural Network Architectures

**Xuefei Ning**[12†*]   **Zixuan Zhou**[1*]   **Junbo Zhao**[1]   **Tianchen Zhao**[1]   **Yiping Deng**[2]

**Changcheng Tang**[3]   **Shuang Liang**[3]   **Huazhong Yang**[1]   **Yu Wang**[1†]

Department of Electronic Engineering, Tsinghua University[1]
TCS Lab, Huawei[2]
Novauto Technology Co. Ltd.[3]

## Abstract

Neural architecture search tries to shift the manual design of neural network (NN) architectures to algorithmic design. In these cases, the NN architecture itself can be viewed as data and needs to be modeled. A better modeling could help explore novel architectures automatically and open the black box of automated architecture design. To this end, this work proposes a new encoding scheme for neural architectures, the Training-Analogous Graph-based ArchiTecture Encoding Scheme (TA-GATES). TA-GATES encodes an NN architecture in a way that is analogous to its training. Extensive experiments demonstrate that the flexibility and discriminative power of TA-GATES lead to better modeling of NN architectures. We expect our methodology of explicitly modeling the NN training process to benefit broader automated deep learning systems. The code is available at `https://github.com/walkerning/aw_nas`.

## 1 Introduction

The past decade has witnessed tremendous advances of deep learning (DL) methods using neural networks (NNs). One of the key driving forces behind NN's widespread applications is the clever design of NN architectures that are both effective and efficient [15, 37, 11, 13]. Nevertheless, long-tail distributed tasks and platforms in the real world pose challenges to the scalability of manual architecture design workflows. One pathway to tackle these challenges is to construct Automated Deep Learning (AutoDL) systems [38, 32, 9] that can decide on factors in all aspects of NN training and inference in an automated and algorithmic way. Neural Architecture Search (NAS) [57, 8, 46] is a subfield of AutoDL that focuses on automated architecture exploration.

As NAS shifts up the architecture design level from manual design to algorithmic design, NN architectures themselves can be viewed as data and need to be modeled. For example, predictor-based NAS [27, 23, 42, 20] constructs a performance predictor consisting of an architecture encoding scheme and a prediction head, and uses its predictions for unseen architectures to guide architecture sampling. A better architecture encoding scheme enables more accurate predictions, and thereby enable us to explore the space more efficiently by only sampling promising architectures [23, 27, 43]. Moreover, an encoding scheme can also be used to improve the widely-used parameter-sharing evaluation [55] or help open the black box of automated architecture design [33]. To this end, our work aims at designing a better encoding scheme for NN architectures.

State-of-the-art (SOTA) encoding schemes [54, 27] of NN architectures are graph-based ones that view an architecture as a directed acyclic graph (DAG) of computations. As shown in Fig. 1 and described

---

*Equal contribution.
†Corresponding to: foxdoraame@gmail.com, yu-wang@tsinghua.edu.cn

36th Conference on Neural Information Processing Systems (NeurIPS 2022).

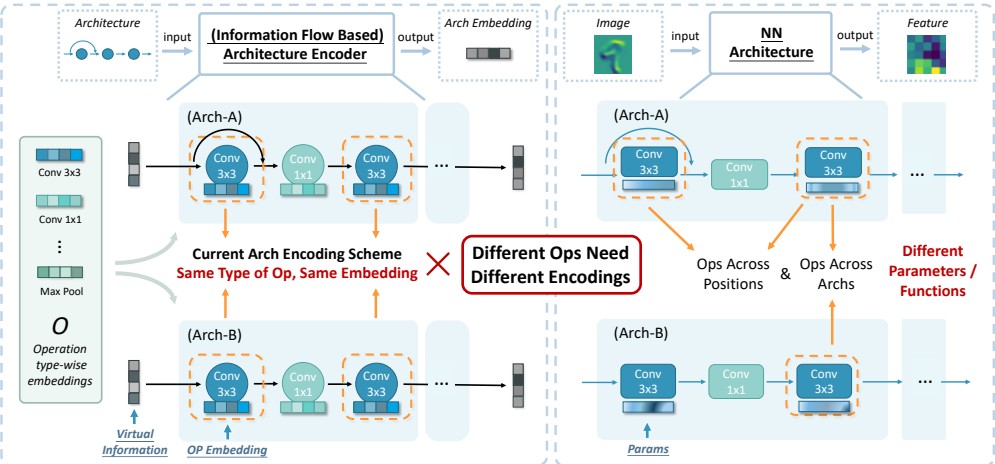

Figure 1: Motivation illustration. **Left**: State-of-the-art information flow based encoders (architecture as input, architecture encoding as output) [54, 27]. In their encoding process, all Conv3x3 share the embedding in $O$, no matter which architecture or which position the operation is in. **Right**: NN architecture (data as input, feature as output). Conv3x3 operations across positions (e.g., the two Conv3x3 in Arch-A) and across architectures (e.g., the first Conv3x3 in Arch-A and Arch-B) obtain different parameters through NN training represent different functions. For example, different from a plain Conv3x3, a Conv3x3 with a surrounding skip connection represents a residual function.

in Sec. A.1, these encoding schemes map each type of computation or operation (e.g., MaxPool, Conv3x3) to the same embedding, and the operation type-wise embeddings $O$ are trainable parameters of the encoder. And their encoding process of an architecture DAG could be seen as a mimicking of how the architecture handles real data. More concretely, in the real NN forward propagation, the images are put at the input node, flow through the architecture, get processed by each operation, and output feature maps at the final output node. While in the encoding process of GATES [27], analogically, the input node embedding flows through the architecture DAG, gets multiplied by the embedding corresponding to each operation, and outputs the architecture encoding at the final output node. This encoding process mimicking the NN forward propagation is shown to outperform non-graph-based encoding schemes [23, 42, 20, 47] and vanilla GCN-based schemes [35, 10].

However, these two schemes neglect an important characteristic of NN architectures: an architecture is not a DAG with fixed computations (e.g., $\sin, \cos$, MaxPool), but a DAG with trainable operations (i.e., parameterized operations, e.g., Conv3x3). That is to say, operations of the same type (e.g., two Conv3x3) can represent different functions since they obtain different parameters through NN training. When encoding an architecture, the SOTA encodings [54, 27] make all operations of the same type across different locations and architectures use the same embedding, which does not take account of this characteristic of architectures. Fig. 1 illustrates that we need a scheme that can give contextualized embeddings for operations according to their architectural context.

To realize this, we base our design on the fundamental idea that an NN architecture not only depicts the computations in a forward propagation, but also implies its learning dynamics. In view of this, we propose an intuitive method, Training-Analogous Graph-based ArchiTecture Encoding Scheme (TA-GATES). During the encoding process of an architecture, TA-GATES mimics the learning dynamics of its parameterized operations. This training-analogous modeling enables TA-GATES to better encode NN architectures. We summarize the contributions of this paper as follows.

- We propose TA-GATES, a novel encoding scheme for NN architectures with analogous modeling of the NN training process. To encode an architecture DAG, TA-GATES conducts an iterative process of forward and backward passes on the DAG for several time steps, and uses the forward output at the final time step as the architecture encoding. In each time step, TA-GATES updates the embeddings of parameterized operations. This encoding process that iteratively updates the operation embeddings could be seen as a mimicking of the architecture training process that iteratively updates the operation parameters. In this way, operations get contextualized embeddings according to their architectural context.

- To further improve the discriminative power of TA-GATES (see Sec. 3.2), we propose symmetry-breaking techniques for operation embeddings at the beginning of the encoding process (before the iterative time steps). This technique could be seen as a mimicking of the random parameter initialization in the actual architecture training process.

- We apply TA-GATES for performance prediction in various architecture benchmarks. Experiments show that TA-GATES consistently surpasses baseline schemes across search spaces, for different tasks (ranking, regression, anytime prediction) and under different settings.

## 2 Related Work

**Automated Deep Learning.**   In the complex pipeline to build DL models, there are many steps and components that require substantial expert knowledge. The need to design DL models for vast tasks and platforms has aroused research interests in AutoDL. NAS [8, 46] is proposed to automatically design the NN architecture. As for other training-time design choices, hyper-parameter optimization (HPO) [38] is a long-standing topic that dates back to 1990s [56]. AutoAug [3, 4] aims at designing data augmentation. And AutoLoss methods [18, 17] automate the design of loss functions.

**Architecture Benchmarks.**   Topological architecture search spaces can be classified into the operation-on-node (OON) ones and the operation-on-edge (OOE) ones. Architectures in OON or OOE spaces have operations on their nodes or edges, respectively. Researchers have established many benchmarks for the ease of comparing NAS methods [51, 52, 7, 30, 36, 50]. Benchmarks on OON spaces include NAS-Bench-101 (NB101) [51], NAS-Bench-1Shot1 [52]. And as for OOE spaces, NAS-Bench-201 (NB201) [7] is a benchmark on a small search space of one cell architecture. NDS ENAS [30] and NAS-Bench-301 (NB301) [36] provide benchmarks on larger search spaces [29, 21] of two cell architecture (i.e., the normal and reduce cells).

**Predictor-based NAS.**   A NAS method consists of two components: (1) The evaluation strategy gives out the performance of an architecture, e.g., by training on the training dataset to get its parameters and then testing on the validation dataset [57]. (2) The search strategy samples architectures to evaluate and explores new architectures according to the feedback of the evaluation strategy.

To improve the exploration efficiency in the large space, the predictor-based search strategy [27, 23, 42, 20] trains an approximate performance predictor of architectures using some pairs of architecture and ground-truth (GT) performance, and uses its prediction scores of unseen architectures to guide architecture sampling. An accurate predictor helps sample architectures more likely to be well-performing and improves NAS results. Lots of efforts have been devoted to developing the architecture encoding scheme, as it is essential for the predictor's generalization ability to unseen architectures.

**Architecture Encoding Schemes.**   Existing encoding schemes of NN architectures include non-graph-based ones and graph-based ones. Non-graph-based schemes convert the computational DAG into a sequence [23, 42, 20, 47, 22, 35, 44] or image [47], and apply XGBoost, Multi-Layer Perceptron (MLP), Long Short-Term Memory (LSTM), or Convolutional Neural Network (CNN). They do not explicitly use the graph topology, and have notable weakness such as improper isomorphism handling.

As for graph-based schemes, there are attempts [35, 10] to apply plain graph convolutional networks (GCNs) [14] to encode architectures. As plain GCNs cannot be applied to OOE spaces directly, to encode architectures in OOE spaces, these methods either adopt the line-graph conversion trick [35, 45] or propose ad-hoc solutions [10] to convert architectures into OON graphs. NASBOWL [33] proposes to use the WL graph kernel with multiple iterations in the Gaussian Process surrogate. These schemes are not dedicatedly designed for NN architectures. SOTA graph-based schemes, D-VAE [54] and GATES [27], view an NN architecture as a computational DAG and follow its "information flow" to encode it. Their analogous modeling of the NN forward propagation provides better encoding.

Another related work, CATE [49], proposes a transformer-based encoder. The motivation behind CATE's transformer design is to capture "deep contextualized information" of operations. They use reachability-masked attention to aggregate operation embeddings and get contextualized operation embeddings. In this work, we propose a more intrinsical way to get contextualized operation embeddings, which can also enable more potentials such as anytime prediction.

# 3 Training-Analogous GATES

State-of-the-art encoding schemes [54, 27] view an NN architecture as a computational DAG on which the information flows and gets processed. We refer to them as the information flow-based schemes. As shown in Fig. 1 (left), in their encoding process, a piece of virtual information (a trainable encoder parameter $E$ in Alg. 1) is used as the input node embedding of the architecture DAG. Then, this information flows along the DAG, and when the information arrives at an operation, the information is transformed according to the embedding of that operation as shown in Equ. A2 and Equ. A3. Finally, the information at the output node is adopted as the architecture embedding (see also Sec. A.1). Their encoding process of an architecture is analogous to its data processing.

We notice that they neglect an important characteristic of NN architectures: An NN architecture is not a DAG with fixed computations, but a DAG with trainable operations. These trainable operations are parametrized, and their parameters are obtained through a training process, which implies that two operations of the same type with different parameters represent different functions.

As discussed in Fig. 1, neglecting this characteristic can result in improper modeling of operations and architectures. Is there an intrinsical way to solve this issue? Based on the fundamental idea that an NN architecture not only depicts the computations in the forward propagation but also implies the learning dynamics, TA-GATES encodes architectures by mimicking their training process.

In the following, Sec. 3.1 first describes the encoding process of TA-GATES. Then, we elaborate on our proposed "symmetry-breaking" technique in Sec. 3.2. And Sec. 3.3 describes how the explicit modeling of the training process can empower the anytime performance prediction task.

## 3.1 Iterative Encoding in Analogy to Iterative Parameter Training

We give out the encoding process of TA-GATES in Alg. 1 (Line 1-10) and an illustration in Fig. 2 (upper). The learnable parameters and modules of TA-GATES and notations are also summarized in the first two sections in Alg. 1. We break down the encoding process as follows. First, we get the initial operation embedding $\text{emb}_{\text{op}}^{\text{ori}} = \text{GetEmbedding}(O, op) \in \mathbb{R}^{M \times d_o}$ for all $M$ operations in the DAG according to their operation types (Line 1). Note that $O \in \mathbb{R}^{N_o \times d_o}$ is an operation type-wise embedding matrix, where each row corresponds to one operation type. At this point, operations of the same type have the same embeddings in $\text{emb}_{\text{op}}^{\text{ori}}$. Then, we apply SymmetryBreaking on $\text{emb}_{\text{op}}^{\text{ori}}$ to get $\text{emb}_{\text{op}}^{(0)}$ (Line 2), which will be discussed in detail in Sec. 3.2.

Then, as shown in Fig. 2 (upper left), we conduct an iterative process of forward and backward passes on the DAG $\alpha$ for $T$ time steps, in which the operation embeddings $\text{emb}_{\text{op}}$ get iteratively updated. This process can be seen as mimicking the iterative parameter updates of the architecture in actual training. In each time step $t$, we first compute the information flow based GCNs on the architecture DAG $a$ (Line 4-5). Specifically, we use a global trainable parameter $E$ as the input information of the architecture DAG (i.e., feed $E$ into the input node of $a$), and call the InfoPropagation procedure with the current operation embedding $\text{emb}_{\text{op}}^{(t-1)}$. InfoPropagation is described in Sec. A.1 and implemented following [27, 54]. After the forward GCN pass, we convert the output information of the forward pass $f_{\text{info}}^{(t)}[N]$ by applying FBConvert (an MLP) to get $b_{\text{info}}^{(t)}[N]$. Then $b_{\text{info}}^{(t)}[N]$ is used as the input information of the backward GCN pass in Line 7. Note that $b_{\text{info}}^{(t)}[N]$ is fed into the output node of $a$, i.e., the input node of $a^T$. We call the two GCN passes in Line 5/7 the forward and backward passes as they propagate on the DAG $a$ and its transposed DAG $a^T$, respectively.

After the forward and backward GCN pass in the $t$-th time step, TA-GATES computes the operation embedding updates using the propagated information in the forward and backward passes (Line 8). Specifically, GetOpEmbUpdate concatenates the operation embedding from the last time step $\text{emb}_{\text{op}}^{(t-1)}$ and the propagated information $f_{\text{info}}^{(t)}, b_{\text{info}}^{(t)}$ in the forward and backward passes as the input, feed it into an MLP, and get the operation embedding updates $\delta^{(t)}$. Then, $\delta^{(t)}$ is updated onto the operation embeddings to get $\text{emb}_{\text{op}}^{(t)}$ (Line 9). And the updated operation embeddings $\text{emb}_{\text{op}}^{(t)}$ will be used in the following time step $t + 1$.

After $T$ time steps, the output of the $T$-th forward InfoPropagation pass (i.e., the node embedding at the output node) $f_{\text{info}}^{(T)}[N]$ is used as the architecture encoding.

---
**Algorithm 1** The Encoding Process of TA-GATES
---

**Learnable Parameter or Modules of TA-GATES:**

    $E$: the input information

    $W^f, W^b$: the parameters of forward and backward GCNs (see Sec. A.1)

    $O \in \mathbb{R}^{N_o \times d_o}$: the embeddings of $N_o$ types of operations

    FBConvert: an MLP module to convert the forward pass' output to the backward pass' input

    GetOpEmbUpdate: an MLP module to get updates of operation embeddings

    InfoPropagation: the information flow based GCN computation (see Sec. A.1)

    SymmetryBreaking: symmetry breaking using parameter-level zero-cost metrics (see Equ. 1)

**Other Notations:**

    $T$: the number of time steps

    $N$: the number of nodes in a cell architecture

    $M$: the number of operations in a cell

    $N_o$: the number of operation choices in the search space (the subscript $o$ denotes operation)

    $d_o$: the dimension of operation embeddings (the subscript $o$ denotes operation)

    $\text{emb}_{\text{op}}^{(t)}$: the operation embeddings at time step $t$

    $f_{\text{info}}^{(t)}[n]$, $b_{\text{info}}^{(t)}[n]$: the information of the $n$-th node in the forward or backward pass at time step $t$
    ($n = 1$ denotes the input node, $n = N$ denotes the output node)

    $s$: a scale of operation embeddings' updates, set to 0.1 in all experiments except ablation studies

**Input:**

    $a$: a DAG denoting the NN architecture, $a^T$ denotes its transposed DAG with all edges reversed
    $op \in \{0, \cdots, N_o\}^M$: the operation indexes in $a$

**Encoding Process:**

  1: $\text{emb}_{\text{op}}^{\text{ori}} = \text{GetEmbedding}(O, op)$

  2: $\text{emb}_{\text{op}}^{(0)} = \text{SymmetryBreaking}(\text{emb}_{\text{op}}^{\text{ori}})$

  3: **for** $t = 1, \cdots, T$ **do**

  4:     $f_{\text{info}}^{(t)}[1] = E$

  5:     $f_{\text{info}}^{(t)}[2 : N] = \text{InfoPropagation}(f_{\text{info}}^{(t)}[1]; a, \text{emb}_{\text{op}}^{(t-1)}, W^f)$

  6:     $b_{\text{info}}^{(t)}[N] = \text{FBConvert}(f_{\text{info}}^{(t)}[N])$

  7:     $b_{\text{info}}^{(t)}[1 : N-1] = \text{InfoPropagation}(b_{\text{info}}^{(t)}[N]; a^T, \text{emb}_{\text{op}}^{(t-1)}, W^b))$

  8:     $\delta^{(t)} = \text{GetOpEmbUpdate}([\, \text{emb}_{\text{op}}^{(t-1)} \mid f_{\text{info}}^{(t)} \mid b_{\text{info}}^{(t)} \,])$

  9:     $\text{emb}_{\text{op}}^{(t)} = \text{emb}_{\text{op}}^{(t-1)} + s \times \delta^{(t)}$

10: **end for**

**Output:** $f_{\text{info}}^{(T)}[N]$    ($\{f_{\text{info}}^{(t)}[N]\}_{t=1,\cdots,T}$ if doing anytime performance prediction)

---

**Analogy to NN training.** The analogy between the training process and the TA-GATES's $T$-step iterative encoding process of an NN architecture is illustrated in Fig. 2 (upper). In each time step of TA-GATES, the forward and backward GCN passes (InfoPropagation) correspond to the NN forward and backward propagation, GetOpEmbUpdate correspond to the gradient calculation w.r.t. parameters, and the updates of operation embeddings correspond to parameter updates.

**Improving the modeling flexibility.** The analogous modeling of NN training adds to the flexibility and discriminability of encoding, as it makes the embedding of each operation adaptive to the overall architecture (i.e., contextualized embedding). Thus, two operations with different architectural contexts can get distinguishable embeddings, even if they are of the same type (e.g., the 0-1/0-2 Conv3x3 in Fig. 2 (upper)), which is more reasonable considering that they represent different functions in the actual NN. In this way, both the operation and architecture encodings become more discriminative, bringing benefits to their downstream usage, i.e., architecture encoding for improving predictor-based NAS and operation encoding for improving parameter-sharing evaluation [55].

**Training of TA-GATES.** We have described the encoding process (i.e., the inference) of TA-GATES. As for the training of TA-GATES's parameters (listed in the 1st section in Alg. 1), we construct a performance predictor consisting of TA-GATES and an MLP head, input the architectures, and use their GT performances as the labels for the predictor outputs $\text{MLP}(f_{\text{info}}^{(T)}[N])$ (see Sec. A.3).

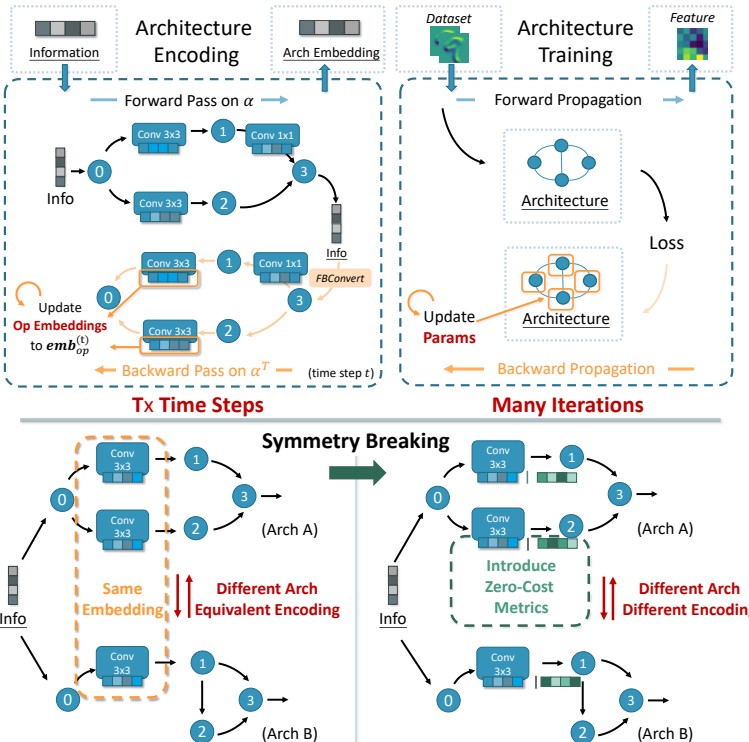

Figure 2: **Upper**: The analogy between the training (Right) of an NN architecture with 4 nodes (0/1/2/3 circles) and **the TA-GATES's iterative encoding process** (Left). **Lower**: A single forward pass in the encoding process *without* (Left) / *with* (Right) **the symmetry-breaking technique**.

## 3.2 Symmetry Breaking in Analogy to Random Parameter Initialization

The iterative encoding process can distinguish operations with different architectural contexts, even if they are of the same type. However, solely with the iterative encoding process, we cannot distinguish operations with fully symmetric architectural contexts and thus fail to discriminate some architectures.

As shown in Fig. 2 (lower left), the two architectures are apparently different, and the upper one has a larger capacity. However, their information-based encodings [54, 27] are indistinguishable. As the three Conv3x3 in Arch-A/Arch-B have the same embeddings, and the operation between node 1 and 2 in Arch-B is a skip connection, node 1 and 2 in Arch-A and Arch-B all have the same information $f_{\text{info}}^{(t)}, b_{\text{info}}^{(t)}$. Therefore, the final architecture encoding of Arch-A and Arch-B is the same. Even if TA-GATES iteratively updates the operation embeddings in the architecture encoding process, the embeddings of these Con3x3 operations remain the same across all time steps if their initial embeddings at time step 0 are the same, since they have exactly the same paths to the input and output nodes (i.e., at fully symmetrical positions).

So, how to enable the encoding scheme to distinguish operations with fully symmetric architectural contexts? Actually, the reason why the Conv3x3s and the architectures in Fig. 2 (lower left) are not equivalent in NN training is that the random parameter initialization breaks the symmetry of the two Conv3x3 when the training begins. Based on this analysis, we propose to apply symmetry breaking to the initial operation embeddings $\text{emb}_{\text{op}}^{(0)}$ (SymmetryBreaking in Alg. 1 Line 2) to enable TA-GATES to distinguish symmetric operations. We propose three types of symmetry-breaking techniques:

1. **Using random noises:** The most straightforward way is to add random noises onto $\text{emb}_{\text{op}}^{(0)}$.
2. **Using zero-cost saliency metrics:** Instead of directly injecting randomness into operation embeddings, we can inject randomness into NN parameters and aggregate some metrics of parameters in each operation to refine its embedding. We propose to aggregate parameter-wise saliency metrics to break the symmetry of operation embeddings. Specifically, SymmetryBreaking in the encoding process of an architecture first randomly initializes the parameters of the architecture. Then, for each operation, we calculate and aggregate five per-parameter saliency metrics as a 5-dim vector $Z \in \mathbb{R}^{M \times 5}$, including grad_norm, snip [16], grasp [41], plain [25],

and synflow [40]. And the initial operation embedding $\text{emb}_{\text{op}}^{(0)} \in \mathbb{R}^{M \times d_o}$ is refined as

$$\text{emb}_{\text{op}}^{(0)} = \text{emb}_{\text{op}}^{\text{ori}} + \beta \times \text{MLP}^z(Z), \tag{1}$$

where the $\text{MLP}^z$ maps from $\mathbb{R}^5$ to $\mathbb{R}^{d_o}$, and $\beta$ is a fixed scale.

3. **Using zero-cost saliency metrics in every time step:** We also experiment with a variant of the 2nd technique: Instead of adding a symmetry-breaking vector onto the initial operation embedding $\text{emb}_{\text{op}}^{\text{ori}}$, we concatenate $Z$ onto $\text{emb}_{\text{op}}^{(t)}$ in all time steps.

Note that though the last two techniques aggregate parameter-level metrics, they only use randomly initialized parameters and do not require actual parameter training. Borrowing the terminology from zero-cost pruning and NAS [40, 1, 24, 19, 26], we refer to them as "zero-cost symmetry-breaking techniques". As illustrated in Fig. 2 (lower right), the encoding with symmetry breaking can differentiate the two operations at symmetric positions in Arch-A, and thereby the architectures Arch-A and Arch-B.

### 3.3 Anytime Performance Training and Prediction

Besides improving the discriminative power of architecture encoding, the explicit modeling of NN training brings other interesting possibilities (see more discussions in Sec. 6). This work explores using TA-GATES to empower anytime performance training and prediction. The meaning of the anytime prediction task is two-fold: (1) Training using performances at other epochs might help improve the prediction of final performances, as these supervisory signals bring more information without inducing additional training costs. (2) The task of predicting multiple performances across epochs has its applications, such as providing inspections into the learning dynamics, or making surrogate benchmarks [50].

The basic strategy to amend existing encoders for anytime prediction is to make it output multiple scores as the predicted performances for multiple epochs. And as it comes to TA-GATES, we have a more natural choice to use output scores of different time steps as the predicted performances, as each time step in TA-GATES corresponds to a checkpoint in the NN training process. We'll demonstrate that this natural fit of TA-GATES indeed boosts the predictive power for anytime performances.

### 3.4 Summary

To summarize, TA-GATES has three key steps in analogy with steps in NN training: (1) Zero-cost symmetry breaking of operation embeddings corresponds to the random initialization of parameters. (2) Forward and backward passes of GCN correspond to the forward and backward propagation of the NN architecture. (3) The updates of operation embeddings correspond to the updates of parameters.

Thanks to the analogous modeling of NN training, TA-GATES has higher flexibility and better discriminative power, and also brings interesting possibilities (see also Sec. 6). For example, TA-GATES provides a more natural and stronger solution. for the anytime performance prediction.

## 4 Experiments

We compare the predictive power of TA-GATES with baseline encoding schemes on four search spaces, including NB101 [51, 52], NB201 [7], NB301 [36], and ENAS [29]. Specifically, we use the performances provided by NAS benchmarks as the ground-truth (GT), and split the GT architecture-performance pairs into training and test splits. After training the predictor with (a subset of) the training split, we measure the predictor fitness on the test split.

Following previous work on architecture encoding, we adopt Kendall's Tau (KD) [34] as the main measure. We also use other measures, including Precision@K (P@K) [27][3], mean squared average error (MSE), and the Pearson coefficient of linear correlation (LC). We use the MSE regression loss to train predictors in anytime training and use the pairwise hinge ranking loss in all other experiments. Detailed experimental settings and runtime information can be found in Sec. B and Sec. C.2, and ablation studies can be found in Sec. C.3.

---

[3]Precision@K: The proportion of true top-K architectures in the predicted top-K architectures.

Table 1: Kendall's Tau of using different encoders on NB101, NB201, NB301 and NDS ENAS. The average result of 9 experiments are reported, and the standard deviation is in the subscript.

| | Encoder | Proportions of 7290 training samples | | | | |
| --- | --- | --- | --- | --- | --- | --- |
| | | 1% | 5% | 10% | 50% | 100% |
| NB101 | MLP [42] | $0.3937_{(0.0302)}$ | $0.5318_{(0.0185)}$ | $0.5703_{(0.0167)}$ | $0.6225_{(0.0078)}$ | $0.6307_{(0.0069)}$ |
| | LSTM [42] | $0.5476_{(0.0341)}$ | $0.5876_{(0.0245)}$ | $0.6040_{(0.0154)}$ | $0.6196_{(0.0142)}$ | $0.6131_{(0.0185)}$ |
| | GCN (global node) [35] | $0.3668_{(0.0563)}$ | $0.5973_{(0.0233)}$ | $0.6927_{(0.0108)}$ | $0.7520_{(0.0075)}$ | $0.7689_{(0.0083)}$ |
| | NASBOWL [33] | $0.5850_{(0.0232)}$ | $0.6416_{(0.0241)}$ | $0.6536_{(0.0193)}$ | $0.6833_{(0.0022)}$ | $0.6872_{(0.0000)}$ |
| | SemiNAS [22] | $0.5273_{(0.0589)}$ | $0.6055_{(0.0294)}$ | $0.5953_{(0.0279)}$ | $0.6040_{(0.0284)}$ | $0.6043_{(0.0179)}$ |
| | XGBoost [44] | $0.4517_{(0.0470)}$ | $0.5987_{(0.0365)}$ | $0.5680_{(0.0125)}$ | $0.5677_{(0.0077)}$ | $0.6175_{(0.0000)}$ |
| | GATES [27] | $0.6321_{(0.0251)}$ | $0.7493_{(0.0166)}$ | $0.7690_{(0.0077)}$ | $0.7999_{(0.0071)}$ | $0.8119_{(0.0071)}$ |
| | TA-GATES | $\mathbf{0.6686}_{(0.0338)}$ | $\mathbf{0.7744}_{(0.0211)}$ | $\mathbf{0.7839}_{(0.0063)}$ | $\mathbf{0.8133}_{(0.0053)}$ | $\mathbf{0.8217}_{(0.0057)}$ |

| | Encoder | Proportions of 7813 training samples | | | | |
| --- | --- | --- | --- | --- | --- | --- |
| | | 0.1% | 0.5% | 1% | 5% | 10% |
| NB201 | MLP [42] | $0.0162_{(0.0859)}$ | $0.0863_{(0.0556)}$ | $0.1756_{(0.0332)}$ | $0.3885_{(0.0237)}$ | $0.5492_{(0.0092)}$ |
| | LSTM [42] | $0.1935_{(0.1806)}$ | $0.5079_{(0.0715)}$ | $0.5691_{(0.0110)}$ | $0.6690_{(0.0189)}$ | $0.7395_{0.0061)}$ |
| | Line-Graph GCN [35] | $0.2461_{0.1549}$ | $0.3113_{0.0626}$ | $0.4080_{0.0369}$ | $0.5461_{0.0138}$ | $0.6095_{0.0164}$ |
| | NASBOWL [33] | $0.4980_{(0.0408)}$ | $0.6674_{(0.0077)}$ | $0.5912_{(0.0874)}$ | $0.7259_{((0.0098)}$ | $0.7625_{(0.0083)}$ |
| | XGBoost [44] | $0.0706_{(0.1238)}$ | $0.3719_{(0.0560)}$ | $0.4178_{(0.0288)}$ | $0.6412_{(0.0053)}$ | $0.7084_{(0.0123)}$ |
| | GATES [27] | $0.4309_{0.1062}$ | $0.6702_{0.0254}$ | $0.7571_{0.0169}$ | $0.8583_{0.0019}$ | $0.8823_{0.0024}$ |
| | TA-GATES | $\mathbf{0.5382}_{(0.0478)}$ | $\mathbf{0.6707}_{(0.0256)}$ | $\mathbf{0.7731}_{(0.0249)}$ | $\mathbf{0.8660}_{(0.0060)}$ | $\mathbf{0.8890}_{(0.0049)}$ |

| | Encoder | Proportions of 5896 training samples | | | | |
| --- | --- | --- | --- | --- | --- | --- |
| | | 0.5% | 1% | 5% | 10% | 50% |
| NB301 | MLP [42] | $0.2750_{(0.0722)}$ | $0.4018_{(0.0209)}$ | $0.5373_{(0.0093)}$ | $0.5687_{(0.0060)}$ | $0.6249_{(0.0021)}$ |
| | LSTM [23] | $0.5161_{(0.0446)}$ | $0.5689_{(0.0218)}$ | $0.6893_{(0.0047)}$ | $0.7144_{(0.0032)}$ | $0.7572_{(0.0019)}$ |
| | GCN [10] | $0.0951_{(0.0350)}$ | $0.1280_{(0.0441)}$ | $0.2673_{(0.0061)}$ | $0.2835_{(0.0059)}$ | $0.3179_{(0.0013)}$ |
| | XGBoost [44] | $0.2725_{(0.0395)}$ | $0.3059_{(0.0285)}$ | $0.3313_{(0.0120)}$ | $0.3227_{(0.0217)}$ | $0.3461_{(0.0034)}$ |
| | GATES [27] | $0.5616_{(0.0251)}$ | $0.6064_{(0.0275)}$ | $0.6916_{(0.0112)}$ | $0.7180_{(0.0067)}$ | $0.7595_{(0.0027)}$ |
| | TA-GATES | $\mathbf{0.5728}_{(0.0307)}$ | $\mathbf{0.6351}_{(0.0138)}$ | $\mathbf{0.7123}_{(0.0087)}$ | $\mathbf{0.7331}_{(0.0071)}$ | $\mathbf{0.7685}_{(0.0066)}$ |

| | Encoder | Proportions of 500 training samples | | | | |
| --- | --- | --- | --- | --- | --- | --- |
| | | 5.0% | 10.0% | 25.0% | 50.0% | 100.0% |
| ENAS | MLP [42] | $0.1607_{(0.0518)}$ | $0.2264_{(0.0470)}$ | $0.3543_{(0.0139)}$ | $0.3858_{(0.0090)}$ | $0.4141_{(0.0036)}$ |
| | LSTM [23] | $0.2594_{(0.0573)}$ | $0.3406_{(0.0370)}$ | $0.4509_{(0.0175)}$ | $0.4875_{(0.0133)}$ | $0.5517_{(0.0048)}$ |
| | GCN [10] | $0.2301_{(0.0923)}$ | $0.3140_{(0.0151)}$ | $0.3367_{(0.0080)}$ | $0.3508_{(0.0108)}$ | $0.3715_{(0.0041)}$ |
| | GATES [27] | $0.3400_{(0.0417)}$ | $0.4286_{(0.0104)}$ | $0.5274_{(0.0245)}$ | $0.5971_{(0.0128)}$ | $0.6467_{(0.0119)}$ |
| | TA-GATES | $\mathbf{0.3458}_{(0.0383)}$ | $\mathbf{0.4407}_{(0.0104)}$ | $\mathbf{0.5485}_{(0.0251)}$ | $\mathbf{0.6324}_{(0.0128)}$ | $\mathbf{0.6683}_{(0.0076)}$ |

## 4.1 Comparison with Baseline Encoders

Table 1 shows the Kendal's Tau on test set using different encoders. Different columns show the results of training using different numbers of architectures. For example, a proportion of 1% on NB101 means that 1%×7290=72 architectures are used for predictor training. The comparison of P@K is shown in Fig. 3. We can see that TA-GATES achieves superior ranking quality consistently. We also conduct architecture search experiments using TA-GATES and discuss the results in Sec. C.2.

## 4.2 Comparison of Symmetry-Breaking Techniques

As shown in Table 2, except for two exceptions when there are only a few training architectures (39/29 training architectures on NB201/NB301), the "Add" symmetry-breaking technique brings improvements on TA-GATES without symmetry breaking ("None"), and achieves the best results. Therefore, we use the "Add" symmetry-breaking technique without explicit statements.

## 4.3 Anytime Performance Training and Prediction

Intuitively, the analogous encoding process of TA-GATES has a natural fit for the anytime performance prediction task. We use MSE training loss to train encoders, and show the KDs and regression measures (LC and MSE) in Table 3, Table A12, Table A13, and Table A14. We can see that TA-GATES significantly boost the anytime prediction performances.

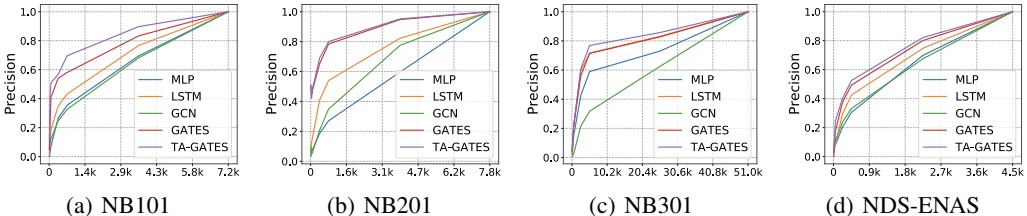

|              | (a) NB101 | (b) NB201 | (c) NB301 | (d) NDS-ENAS |

Figure 3: Precision@K comparison on the validation split of four benchmarks. X-axis: K; Y-axis: Precision. The training proportion is 5% on NB101, NB201, and NB301, and 50% on NDS ENAS.

Table 2: Kendall's Tau of using different symmetry-breaking techniques. "None" indicates TA-GATES without symmetry breaking. "Random", "Add", "Concat" refer to TA-GATES with the three symmetry-breaking techniques described in Sec. 3.2, respectively.

| Method | NB101 (7290 training) | | | | NB201 (7812 training) | | | | NB301 (5896 training) | | | |
|--------|------|------|------|------|------|------|------|------|------|------|------|------|
|        | 1%   | 5%   | 10%  | 50%  | 0.5% | 1%   | 5%   | 10%  | 0.5% | 1%   | 5%   | 10%  |
| GATES  | 0.6321 | 0.7493 | 0.7690 | 0.7999 | 0.6702 | 0.7571 | 0.8583 | 0.8823 | 0.5616 | 0.6064 | 0.6916 | 0.7180 |
| None   | 0.6510 | 0.7581 | 0.7704 | 0.8070 | 0.6838 | 0.7667 | 0.8623 | 0.8866 | 0.5735 | 0.6182 | 0.7020 | 0.7280 |
| Random | 0.6425 | 0.7612 | 0.7711 | 0.8020 | 0.6777 | 0.7688 | 0.8634 | 0.8836 | **0.5735** | 0.6207 | 0.7034 | 0.7257 |
| Concat | 0.6585 | 0.7689 | 0.7819 | 0.8086 | **0.6847** | 0.7708 | 0.8632 | 0.8856 | 0.5659 | 0.6230 | 0.7048 | 0.7261 |
| Add    | **0.6686** | **0.7744** | **0.7839** | **0.8133** | 0.6707 | **0.7731** | **0.8660** | **0.8890** | 0.5728 | **0.6351** | **0.7123** | **0.7331** |

Table 3: Kendall's Tau of anytime training and prediction. **Upper/Lower**: Kendall's Tau with the half GT accuracy in the middle of training / the final GT accuracy. **Baselines**: "Single-" means to only use one supervisory signal to train a predictor. "Multi-" refers to basic strategy described in Sec. 3.3: The predictor outputs multiple scores and is trained with multiple supervisory signals.

| KD with the half accuracy | | NB101 (7290 training) | | | | NB201 (7812 training) | | | | NB301 (5896 training) | | | |
|---------|-----------|------|------|------|------|------|------|------|------|------|------|------|------|
| Encoder | Training  | 1%   | 5%   | 10%  | 50%  | 0.5% | 1%   | 5%   | 10%  | 0.5% | 1%   | 5%   | 10%  |
| Single-GATES | half | 0.3636 | 0.3473 | 0.3147 | 0.4796 | 0.5882 | 0.6654 | 0.7317 | 0.7732 | 0.2028 | 0.2106 | 0.2807 | 0.3347 |
| Multi-LSTM | half+final | 0.0123 | 0.0659 | 0.0723 | 0.0518 | 0.3189 | 0.3797 | 0.4771 | 0.5285 | 0.1914 | 0.1464 | 0.1132 | 0.1187 |
| Multi-GATES | half+final | 0.2862 | 0.2912 | 0.2883 | 0.1413 | 0.5827 | 0.6574 | 0.7168 | 0.7745 | 0.1635 | 0.1654 | 0.1481 | 0.1263 |
| TA-GATES | half+final | **0.3921** | **0.4615** | **0.4805** | **0.5674** | **0.6297** | **0.7110** | **0.7827** | **0.8140** | **0.2345** | **0.2322** | **0.3092** | **0.4249** |
| **KD with the final accuracy** | | NB101 (7290 training) | | | | NB201 (7812 training) | | | | NB301 (5896 training) | | | |
| Encoder | Training  | 1%   | 5%   | 10%  | 50%  | 0.5% | 1%   | 5%   | 10%  | 0.5% | 1%   | 5%   | 10%  |
| Single-GATES | final | 0.3856 | 0.3820 | 0.5034 | 0.5903 | 0.4914 | 0.6915 | 0.7237 | 0.7806 | 0.1557 | 0.1549 | 0.1954 | 0.2432 |
| Multi-LSTM | half+final | -0.0372 | 0.1028 | 0.2191 | 0.1473 | 0.4166 | 0.4795 | 0.5491 | 0.6062 | **0.2046** | 0.2153 | 0.2283 | 0.2318 |
| Multi-GATES | half+final | 0.3455 | 0.3341 | 0.3370 | 0.1818 | 0.6145 | 0.6902 | 0.7306 | 0.7989 | 0.1190 | 0.1209 | 0.1025 | 0.0868 |
| TA-GATES | half+final | **0.5463** | **0.5850** | **0.5950** | **0.6477** | **0.6648** | **0.7363** | **0.8213** | **0.8624** | 0.1963 | **0.2944** | **0.2928** | **0.4178** |

On one hand, TA-GATES enables the predictor to benefit from multiple supervisory signals by capturing the learning speed of architectures. Fig. A11 shows that a small architecture Arch-1-A learns faster (higher half accuracy) but ends at a low final accuracy, and a larger architecture Arch-1-B learns slower (lower half accuracy) but ends at a higher accuracy. TA-GATES can correctly predict the relative order of these two architectures both at the end or in the middle of the training process. In contrast, without explicit modeling of the learning dynamics, Multi-GATES tends to give out the same relative order for the half and final accuracies, and thus fails to make correct comparisons.

On the other hand, directly training baseline encoders with half and final accuracies simultaneously even leads to performance degradation. We observe a "trade-off" phenomenon where the prediction fitness for the half and final accuracy have opposite trends. See Sec. C.5 for detailed analyses.

Note that although our anytime training experiments use 2-step TA-GATES (T=2, t=1 for half and t=2 for final) and use the GT accuracies of two training epochs (half and final) for training, TA-GATES can be easily extended to anytime training with more time steps and more supervisory signals.

## 5   Conclusions

Neural architectures are data-processing DAGs with *trainable* operations. According to this nature, this work dedicatedly designs a *Training-Analogous* Graph-based ArchiTecture Encoding Scheme

(TA-GATES) for encoding NN architectures. The encoding process of TA-GATES mimics not only how the information is propagated and processed by operations during the NN inference process, but also the learning dynamics of operations during the NN training process. In this way, every operation becomes distinguishable, i.e., gets "contextualized" embeddings according to its architectural context. Extensive experiments in various search spaces show that TA-GATES consistently improves the performance predictions. We also show how the explicit modeling of NN training in TA-GATES empowers the anytime performance prediction task.

## 6 Broader Applications, Limitations, and Future Directions

Here we discuss two applications that can potentially benefit from the "training-analogous" encoding methodology and one future extension direction of TA-GATES.

**TA-GATES as the learning curve extrapolator for early-stop NAS.** Besides predicting the anytime learning curve from an architecture description, TA-GATES also has the potential to be used as a black-box and learnable extrapolator of partial learning curves. An extrapolator can be handy for accelerating AutoML, where some NN training processes can be early stopped according to the extrapolated estimation at targeting epochs.

The existing extrapolator [6] assumes the NN learning curve can be described by some parametric function families. And during the training process of an architecture, they fit the function family parameters using the validation performances of the early learning curve, and then extrapolate to get the estimations at targeting epochs. Different from existing extrapolators, TA-GATES can take the architecture description as input. That is to say, during the extrapolation, instead of solely relying on the early learning curve, this learnable extrapolator can utilize the learning curve knowledge of other architectures, and thereby has the potential to make better extrapolations in a data-driven way.

**TA-GATES as the joint encoder for other DL factors in AutoML.** As TA-GATES explicitly mimics the NN training process, it is easy to extend TA-GATES to model other training-time factors besides the architecture in an elegant way. Thus, we expect TA-GATES to find its wide application in joint AutoDL systems, not restricting to NAS.

Here list some possibilities of integrating other factors into TA-GATES: (1) Training-time auxiliary towers [39] or architectural reparametrization [5] can be seen as introducing additional nodes and edges in the GCN passes when $t < T$, while still using the inference-time architecture when $t = T$. (2) Different data augmentation can be modeled as different conversions of $E$. (3) Different loss functions can be modeled as different FBConvert. (4) The influence of training hyper-parameters can be modeled by incorporating them into appropriate places in the encoding process. For example, the learning rate schedule is analogous to a schedule of $s$. In a word, the methodology of TA-GATES (i.e., analogous black-box modeling of the NN training process) can empower joint AutoDL systems by combining information regarding both the inference-time and training-time factors in an intrinsic way. TA-GATES could be used in joint AutoDL systems to help explore the inference-time architecture (NAS), the training-time architecture, the data augmentation (AutoAug), the loss function parameters (AutoLoss), hyper-parameters (HPO), and other factors in the complex DL pipeline.

**Enabling TA-GATES to conduct cross-space comparisons.** Although TA-GATES can be applied for different search spaces, it's still limited to in-space comparisons. This is because TA-GATES (1) takes the cell DAG instead of the overall architecture DAG as the input and (2) encodes only the variable choices defined by the space instead of all operation properties. Applying the training-analogous methodology of TA-GATES to develop a universal cross-space predictor [12] is an interesting future work, which can bridge the gap between zero-cost predictors (can conduct cross-space comparisons, non-satisfying predictions) and data-driven but space-specific predictors (cannot conduct cross-space comparisons, have stronger predictive power).

## Acknowledgements

This work was supported by National Natural Science Foundation of China (No. U19B2019, 62171313, 61832007), Beijing National Research Center for Information Science and Technology (BNRist), Tsinghua EE Xilinx AI Research Fund, and Beijing Innovation Center for Future Chips.

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
