# A Implementation Details and Discussions on TA-GATES

## A.1 Implementation of the Information Flow Based GCN

The state-of-the-art information flow based GCN, GATES [27], puts a virtual information of dimension $d_i$ at the input node of a cell architecture, and then calculates the information of other nodes following the topological order. And the computation when encountering an unary operation of type $op \in \{0, \cdots, N_o\}$ goes as follows. A soft attention mask $m \in (0, 1)^{d_i}$ is firstly computed as

$$m = \sigma(\text{GetEmbedding}(O, op) \ W_o), \tag{A2}$$

where $\sigma$ is the sigmoid function. And $O \in \mathbb{R}^{N_o \times d_o}$ is the embeddings of $N_o$ types of operations, where $d_o$ denotes the dimension of each operation embedding. $\text{GetOpEmbedding}(O, o)$ get out the embedding for this type of operation and has a dimension of $d_o$, and $W_o \in \mathbb{R}^{d_o \times d_i}$ is a linear transformation matrix.

Then, the output information of this unary operation is calculated as

$$x_{\text{out}} = m \odot x_{\text{in}} W_x, \tag{A3}$$

where $\odot$ denotes the elementwise multiplication. And a summation is used to aggregate virtual information on multiple incoming edges of a node.[4] Finally, the information at the output node of the cell architecture is used as the architecture embedding. We also give a graphical illustration of the GATES encoding process in Fig. A4.

Note that we cannot directly take the results from the GATES [27] paper, since we conduct experiments on more benchmark search spaces, and we run each of our training experiments with 9 runs (3 different training seeds, and 3 different training sets). Therefore, we use their published codes and rerun all experiments for the baseline encoders.

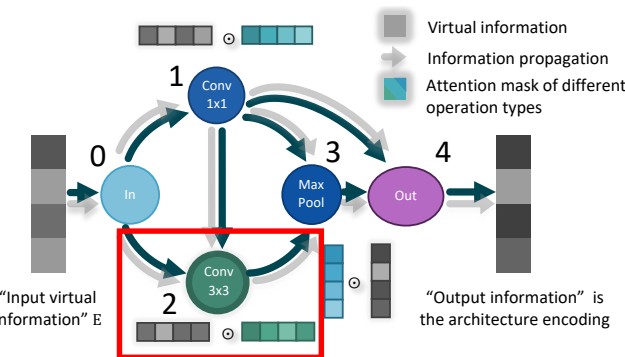

Figure A4: The encoding process of GATES [27] mimics the NN forward process. In the NN forward process, an image is taken as the input data, and each operation processes the data. In the encoding process of GATES, a piece of "virtual information" is taken as the input node embedding, and each operation is a transformation of the propagated information. For example, in the red box, the computation of the feature map $F_2$ at node 2 in the NN forward process is $F_2 = \text{Conv3x3}(F_0 + F_1)$. Analogically, the transformation of the "virtual information" in the encoding process at node 2 is $N_2 = m_2 \odot (N_0 + N_1)$, where the attention mask $m_2 = \sigma(\text{GetEmbedding}(O, \text{Conv3x3})W_o)$ is the attention mask corresponding to the operation type Conv3x3.

**Special handling of skip connections.** We add special handling of all skip connection operations into GATES and TA-GATES. Specifically, their operation embeddings are not treated as learnable parameters during the training process of TA-GATES, and their soft attention masks $m$ are set to all 1s during the encoding process.

This handling helps identify isomorphic architectures caused by skip connections. To be more specific, let us consider the case if a node has one and only one incoming edge, on which the operation is a skip

---

[4]Please refer to [54, 27] for detailed discussions on how they model unary operations and aggregations.

connection. Then, this node is equivalent to its only predecessor node. This type of equivalence results in isomorphic architectures having different original DAG representations. Our special handling of the skip connection ensures the information on these two nodes are equivalent, and thus properly map isomorphic architectures caused by skip connections to the same representation. While without our special handling of the skip connections, the baseline [27] cannot guarantee to map this type of isomorphic architecture pairs to the same representation.

We empirically find that this special handling of skip connections indeed brings slight improvements on GATES [27]. Note that all our reported results of the GATES baseline have already incorporated this improvement by adding this special handling into their published codes.

In some search spaces (such as DARTS [21], NB301 [36], and ENAS [29]), the final output node in a cell architecture conducts a concatenation of some intermediate nodes. DARTS and NB301 concatenate all four intermediate nodes, while ENAS concatenates the loose-end nodes that are not used as the input for other nodes. When preparing the adjacent matrices of these cell architectures, we put "skip connection" operations on the edges between the final output node and its inputs. These skip connections are handled in the same way as the skip connections between other nodes.

### A.2 Implementation of TA-GATES

In this section, we first discuss on several design choices of TA-GATES, i.e., (1) Modeling each operation as a multiplicative transform of the information propagating on graph. (2) Sharing one set of embeddings for operation types, but using different mapping $W_o$ in the forward and backward pass. (3) The input to the GetOpEmbUpdate module. Then, we discuss how TA-GATES handles search spaces with different properties (e.g., operation on node or edge, multi-cell, multi-input-node). Note that these designs are general and correspond to search space properties, instead of being per-search-space designs.

**Discussion on using GATES in the backward information propagation.** In the design of TA-GATES, the backward information propagation on DAG should be analogous to the NN backpropagation. According to the multiplicative chain rule of derivatives in backpropagation, we consider that each operation plays a multiplicative role in NN backpropagation. Thus, we regard using the multiplicative GATES in the backward information propagation to be a reasonable choice, and use it in both the forward and backward information propagation on DAG.

Note that although the forward and backward information passes use the same set of operation embeddings $\text{emb}_{\text{op}}^{(t)}$ in the time step $t$, an operation has different transformations of information in the forward and backward passes. This is because the matrix $W_o$ in Equ. A2 that maps the operation embedding to the attention mask $m$ is different ($W_o^f$ for the forward pass, $W_o^b$ for the backward pass).

**Discussion on GetOpEmbUpdate.** In the NN training process, the parameter updates of each operation are influenced by other operations and the architecture topology. Thus in the encoding process, we need to input the forward and backward information into GetOpEmbUpdate to model the influence. This is why we concatenate $f_{\text{info}}^{(t)}, b_{\text{info}}^{(t)}$ together with $\text{emb}_{\text{op}}^{(t-1)}$ as the input for GetOpEmbUpdate to get $\delta^{(t)}$ in Alg. 1 Line 8.

**Handling different types of search spaces.** Generally speaking, there are two types of topological search spaces, operation-on-node (OON) ones [51, 52], and operation-on-edge (OOE) ones [21, 36, 29]. For completeness, we illustrate an OON and an OOE architecture in Fig. A5[5]

TA-GATES conducts the information propagation passes with the InfoPropogation procedure, which adopts GATES [27] as the propagating method. As GATES is general to OON and OOE search space, we only discuss how to make other procedures general to the OON and OOE search spaces.

The procedures GetEmbedding, SymmetryBreaking, FBConvert, and the operation updates (Line 9) illustrated in Alg. 1 do not need any modifications to handle these two search spaces. As for the procedure GetOpEmbUpdate, for simplicity of expression, Line 8 in Alg. 1 assumes the search space is OON. Concretely, the operation embedding $\text{emb}_{\text{op}}^{(t-1)}$, its corresponding node information in the forward and backward passes are concatenated to feed into GetOpEmbUpdate. And on OOE search

---

[5]We follow [27] to plot this illustration.

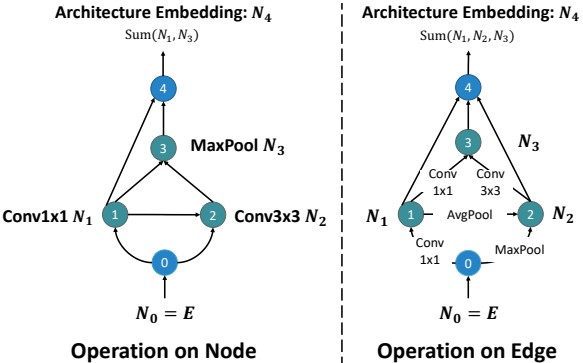

Figure A5: An illustration of example OON (left) and OOE architectures (right).

spaces, denoting the successor node of the edge $e$ on DAG $a$ as $\text{Successor}(e, a)$, the embedding update for an operation on edge $e$ is:

$$\delta^{(t)} = \text{GetOpEmbUpdate}([\, \text{emb}'_{\text{op}} \mid f^{(t)}_{\text{info}}[\text{Successor}(e, a)] \mid b^{(t)}_{\text{info}}[\text{Successor}(e, a^T)]\,]) \qquad \text{(A4)}$$

where $\text{emb}'_{\text{op}}$ is the embedding of this operation in the last time step. In a word, we concatenate the information of the successor node in forward and backward passes with the operation embedding and feed the result into GetOpEmbUpdate to get the embedding update.

**Handling architectures described by multiple cells.**   An architecture in DARTS, NB301, and ENAS search spaces is described by two cell architectures (the normal cell, the reduce cell). Alg. 1 describes the encoding process for one cell architecture. And to encode an architecture described by multiple cells, in every time step, the FBConvert procedure concatenates the output information ($f_{\text{info}}[N]$) of all cell architectures belonging to this architecture, and outputs two separate $b^{(t)}_{\text{info}}[N]$ as the input information for the backward GCN pass of the two cell architectures. The final architecture embedding is obtained by concatenating $f^{(T)}_{\text{info}}[N]$ of consisting cell architectures. All other TA-GATES parameters are shared when encoding different cell architectures.

In a word, FBConnect fuses $f\_\text{info}^{(t)}[N]$ of multiple cell architectures, and then slice it to feed into the backward InfoPropagation. In this way, the backward information and thereby the operation embedding updates in each cell architecture are related to all cell architectures, which is a more reasonable choice than independently encoding each cell architecture.

**Handling multiple input nodes.**   For the ease of understanding, Alg. 1 only shows the case when the number of input nodes of a cell architecture equals 1. When there are more input nodes (as in DARTS, NB301, and ENAS), one just needs to increase the dimension of the learnable $E$ to match the input node number, and initialize the information of the input nodes using $E$ accordingly.

### A.3   Training of TA-GATES

TA-GATES uses the information at the DAG output node in the last time step $f^{(T)}_{\text{info}}[N]$ as the architecture embedding. In the performance prediction task, this architecture embedding is fed into an MLP to predict a score $s$ of this architecture. Denote $\text{E}(a)$ as the information embedding $f^{(T)}_{\text{info}}[N]$ of the architecture $a$, we construct the performance predictor $F$ with a following MLP:

$$s = \text{F}(a) = \text{MLP}(\text{E}(a)) \qquad \text{(A5)}$$

Following [27], we adopt a hinge pair-wise ranking loss to train the predictor.

$$L(\{a_i, y_i\}_N) = \sum_{i=1}^{N} \sum_{j=i+1}^{N} \max(0, m - (\text{F}(a_i) - \text{F}(a_j))) \qquad \text{(A6)}$$

where $a_i$ denotes one architecture, $y_i$ denotes the corresponding ground-truth performance. $m$ denotes the compare margin.

In the anytime performance prediction task, we use the architecture embeddings in other time step $t \neq T$ to predict other performances besides the performance at the final training epoch of the architecture. Denote $\mathrm{E}^{(t)}(a)$ as the information embedding $f_{\mathrm{info}}^{(t)}[N]$ of the architecture $a$. the predicted anytime scores $\{s^{(t)}\}_{t=1,\cdots,T}$ are computed as

$$s^{(t)} = \mathrm{F}^{(t)}(a) = \mathrm{MLP}^{(t)}(\mathrm{E}^{(t)}(a)) \tag{A7}$$

In anytime training, we minimize the Mean Squared Error (MSE) between the predicted and GT performances.

$$L(\{a_i, y_i^{(0)}, y_i^{(1)}, ..., y_i^{(T)}\}_N) = \sum_{i=1}^{N} \sum_{t=1}^{T} (\mathrm{F}^{(t)}(a_i) - y_i^{(t)})^2 \tag{A8}$$

where $y_i^{(t)}$ denotes the GT performance of the architecture $a_i$ corresponding to time step $t$. For example, our experiments set $T$ to be 2, then, $t=1$ and $t=2$ correspond to the half and final GT accuracies of the architecture during its training process, respectively.

## B  Experimental Settings

### B.1  Benchmark Split and Experiment Seeds

For the NB101 search space, we use the 14580 architecture-performance pairs provided by the 3rd subset in NAS-Bench-1shot1 [52]. For each benchmark, we split the architecture-performance pairs into the training split and the test split. The training and validation split sizes are 7290 and 7290 on NB101, 7812 and 7812 on NB201, 5896 and 51072 on NB301, and 500 and 4500 on NDS ENAS. Note that for the surrogate benchmark NB301, we only use the anchor architecture-performance pairs that are obtained through actual training and testing instead of surrogate predictions.

Denoting the size of the overall training split as $|D_t|$, given a training ratio $r$, 9 predictor training experiments are conducted for each encoder configuration (3 training seeds, and 3 different training set). The 3 training seeds are 21, 2021, 202121. As for the 3 different training sets, we first random shuffle the overall training split with seed 2021 and 202121, then for each training ratio $r$, we pick out the first $r|D_t|$ architecture-performance pairs from the original and the two shuffled training splits to train the predictor. In a word, we train the predictor on three training sets with three different training seeds, and average the measures (i.e., KD, LC, and MSE) of the last five epochs. The average result of these 9 experiments are reported.

### B.2  Training of Encoders

We run all predictor training and inference experiments using NVIDIA RTX 3090 GPU and AMD EPYC 7H12. For training with the ranking loss, we adopt a hinge pair-wise ranking loss with margin $m$=0.1, following previous studies [27, 36]. Each predictor is trained using an ADAM optimizer with a learning rate of 1e-3 and a batch size of 512 for 200 epochs.

When training with the regression loss for the anytime performance prediction task, we adopt the MSE regression loss. Each predictor is trained using an ADAM optimizer. The learning rate is set to 1e-3 on NB201 and NDS ENAS, and 1e-4 on NB101 and NB301. The number of training epochs is set to 500 on NB201 and 200 on other benchmarks. The batch size is set to 512 on all benchmarks.

We note that all training configurations are kept the same for TA-GATES and other baseline encoders.

### B.3  Construction of Encoders

On NB101 and NB201, we stack five 128-dim GCN and GATES layers to construct the GCN and GATES encoders, respecitvely. Both the embedding size of the input information $E$ and the operation embedding size are set to 48. For the GCN encoder, we follow the previous study [35] to use the feature of the global node as the architecture embedding, and adopt the line-graph conversion trick [45] to encode architectures on NB201. For TA-GATES, we construct five-layer GATES of dimension 128 for both the forward and backward propagation passes. The information and operation embedding sizes are both set to 48. The FBConvert is constructed by two 128-dim fully-connected layers. Finally, the 128-dim feature of the output node is used as the architecture embedding.

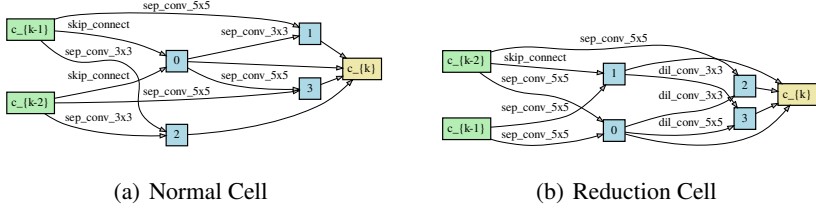

(a) Normal Cell     (b) Reduction Cell

Figure A6: The discovered cell architectures by TA-GATES on NB301.

As for the MLP and LSTM baselines on NB101 and NB201, we construct the MLP encoder by 4 fully-connected layers with 512, 2048, 2048, and 512 nodes, and use the 512-dim output as the architecture's embedding. For the LSTM encoder, the embedding and hidden sizes are both set to 100, and the final hidden state is used as the architecture's embedding. We adopt the serialized representation of the architecture as the input of the MLP and LSTM encoder on NB101, following the previous study [42]. While on NB201, we use the 6 elements in the lower triangular matrix (excluding the diagonal ones) as the input.

On NB301 and ENAS, we concatenate the node and operation lists as the input of MLP and LSTM encoders following [23]. Specifically, the MLP encoder is constructed by three 128-dim fully-connected layers, and the output dimension is set to 32. For the LSTM encoder, we set the operation embedding size to 48 and the hidden size to 128. The final hidden state is used as the embedding. For GCN [10] and GATES [27], we construct the encoders by stacking 64-dim GCN and GATES layers, respectively. All the embedding sizes are set to 32. For TA-GATES, we stack 64-dim GATES layers for both the forward and backward GCN propagation. The number of GCN and GATES layer in GCN, GATES, and TA-GATES is set to 5 on NB301 and 6 on ENAS. The number of layers is set according to the maximum path length in cell architectures (5 on NB301, and 6 on ENAS). The information and operation embedding sizes are set to 32, and the FBConvert is constructed by two 64-dim fully-connected layers.

For the other baselines, we use their original codes to construct the encoders and still use the same training settings as TA-GATES. Specifically, we use NASBOWL code provided by [33] on NB101 and NB201, and use XGBoost code provided by [44] on NB101, NB201 and NB301. As for SemiNAS [22], SemiNAS actually uses an LSTM encoder. As our work focuses on the encoder design and TA-GATES should be orthogonal with the training method, we adopt their encoder construction setting using our LSTM implementation and train with our supervised training setting.

We note that we do not conduct any hyper-parameter search for TA-GATES. Only the number of TA-GATES layers is set to match the longest path in the search space. As for other configurations (e.g., number of hidden units, embedding sizes), we just set them as the same as the baseline GATES.

## C    Additional Experiments and Results

### C.1    More Comparison Results with Baseline Encoders

Table A4 shows the Kendall's Tau with sufficient training data (i.e., 50% and 100% on NB201 and NB301) using different encoders. We can see that TA-GATES also achieves higher ranking quality under this situation.

Fig. A7 shows the comparison results of P@K with more baselines on NB101, NB201, and NB301. We can see that TA-GATES still outperforms these baseline encoders. Besides, Fig. A8 shows the comparison results of P@K with sufficient training data on NB101 and NB301. The results reveal that TA-GATES can achieve better ranking quality consistently.

### C.2    Architectures Search Using TA-GATES

**Architecture Search on NAS Benchmarks**    We conduct predictor-based architecture search on three NAS benchmarks (i.e., NB101, NB201, and NB301) with different architecture encoders. After predictor training, we randomly sample 200 architectures from the search space and select

Table A4: Kendall's Tau of using different encoders on NB201 and NB301 with sufficient training data. The average result of 9 experiments are reported, and the standard deviation is in the subscript.

| Method | NB201 (7812 training) | | NB301 (5896 training) | |
|---|---|---|---|---|
| | 50% | 100% | 50% | 100% |
| MLP [42] | $0.8205_{(0.0050)}$ | $0.8733_{(0.0011)}$ | $0.6249_{(0.0021)}$ | $0.6501_{(0.0014)}$ |
| LSTM [42] | $0.8757_{(0.0018)}$ | $0.9008_{(0.0013)}$ | $0.7572_{0.0019)}$ | $0.7672_{(0.0009)}$ |
| GCN [35] | $0.7733_{0.0000)}$ | $0.8257_{0.0000)}$ | $0.3179_{0.0013)}$ | $0.3256_{0.0016)}$ |
| XGBoost [44] | $0.7977_{(0.0048)}$ | $0.8312_{(0.0000)}$ | $0.3461_{(0.0034)}$ | $0.3766_{(0.0000)}$ |
| GATES [27] | $0.9155_{0.0090)}$ | $\mathbf{0.9259}_{0.0013)}$ | $0.7595_{0.0027)}$ | $0.7670_{0.0053)}$ |
| TA-GATES | $\mathbf{0.9181}_{(0.0041)}$ | $0.9228_{(0.0041)}$ | $\mathbf{0.7685}_{(0.0066)}$ | $\mathbf{0.7766}_{(0.0033)}$ |

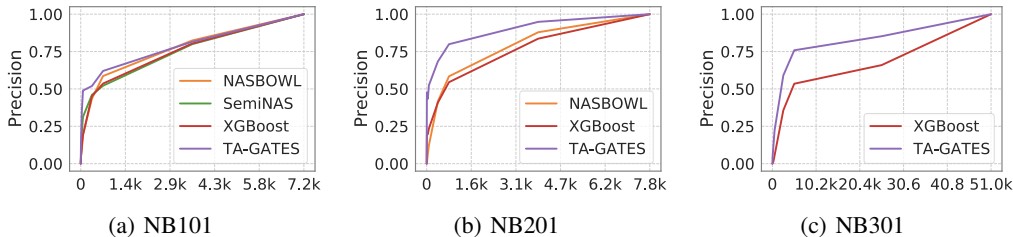

(a) NB101      (b) NB201      (c) NB301

Figure A7: Precision@K comparison on the validation split of NB101, NB201 and NB301. X-axis: K; Y-axis: Precision. The training proportions is 1% on the three benchmarks.

the architecture with the highest predicted score. As shown in Table A5, our proposed method TA-GATES helps to discover the architectures with higher accuracy on all three benchmarks.

Table A5: Accuracy of the discovered architectures on NB101, NB201, and NB301 with different encoders.

| Encoder | NB101 (39 training) | NB101 (79 training) | NB101 (396 training) | NB101 (7920 training) | NB201 (78 training) | NB301 (5 training) | NB301 (58 training) | NB301 (2948 training) | NB301 (5896 training) |
|---|---|---|---|---|---|---|---|---|---|
| MLP | $0.9280_{(0.0142)}$ | $0.9330_{(0.0104)}$ | $0.9402_{(0.0027)}$ | $0.9402_{(0.0027)}$ | $0.8820_{(0.0736)}$ | $0.9374_{(0.0071)}$ | $0.9442_{(0.0021)}$ | $0.9448_{(0.0017)}$ | $0.9447_{(0.0017)}$ |
| LSTM | $0.9359_{(0.0067)}$ | $0.9388_{(0.0052)}$ | $0.9416_{(0.0024)}$ | $0.9434_{(0.0027)}$ | $0.9334_{(0.0045)}$ | $0.9380_{(0.0034)}$ | $0.9440_{(0.0022)}$ | $0.9444_{(0.0019)}$ | $0.9442_{(0.0015)}$ |
| GCN | $0.8750_{(0.1147)}$ | $0.9026_{(0.1119)}$ | $0.9417_{(0.0025)}$ | $0.9434_{(0.0025)}$ | $0.9244_{(0.0151)}$ | $0.9254_{(0.0127)}$ | $0.9214_{(0.0185)}$ | $0.9384_{(0.0053)}$ | $0.9378_{(0.0056)}$ |
| GATES | $0.9407_{(0.0027)}$ | $0.9420_{(0.0025)}$ | $0.9417_{(0.0014)}$ | $0.9428_{(0.0017)}$ | $0.9349_{(0.0036)}$ | $0.9398_{(0.0045)}$ | $0.9454_{(0.0538)}$ | $0.9450_{(0.0014)}$ | $0.9448_{(0.0017)}$ |
| TA-GATES | $\mathbf{0.9424}_{(0.0024)}$ | $\mathbf{0.9426}_{(0.0022)}$ | $\mathbf{0.9429}_{(0.0020)}$ | $\mathbf{0.9443}_{(0.0024)}$ | $\mathbf{0.9393}_{(0.0039)}$ | $\mathbf{0.9417}_{(0.0045)}$ | $\mathbf{0.9466}_{(0.0011)}$ | $\mathbf{0.9450}_{(0.0016)}$ | $\mathbf{0.9452}_{(0.0018)}$ |

**Transferring the Picked Architecture to ImageNet.** We train TA-GATES on NB301 with 10% (589) architectures of the training split, and use it to predict the scores of 10k randomly sampled architectures. We obtain the top 200 architectures based on the predicted scores, and show the architecture among them with the highest NB301 accuracy in Fig. A6.

We stack this cell architecture 14 times to construct the overall network and train it on ImageNet. The initial channel number is set to 48. We train the network for 300 epochs with batch size 256. An SGD optimizer with momentum 0.9 and weight decay 3e-5 is adopted. The learning rate is decayed from 0.1 to 0 following a cosine schedule. In the training process, the gradient norm is clipped to be less than 5, and the dropout rate is set to 0. Following previous work [58, 21], we adopt cutout augmentation with length 16 and the auxiliary tower with a weight of 0.4.

Table A6 compares the test errors of different architectures on ImageNet. As can be seen, the architecture found by TA-GATES can achieve a competitive top-1 error of 24.1% on the test set.

**Discussion on the Training and Inference Time of TA-GATES.** TA-GATES is more complex than GATES. Consequently, the training and inference of TA-GATES are slightly slower than GATES: On NB101, 200-epoch training of TA-GATES and GATES takes 121s and 91s (1% training architectures). Batch testing 7290 architectures with batch size 128 is very fast and only takes about 1s with both encoders. However, these costs are nearly negligible in predictor-based NAS compared to the NN training costs to get the GT accuracies of architectures (>several GPU hours per architecture),

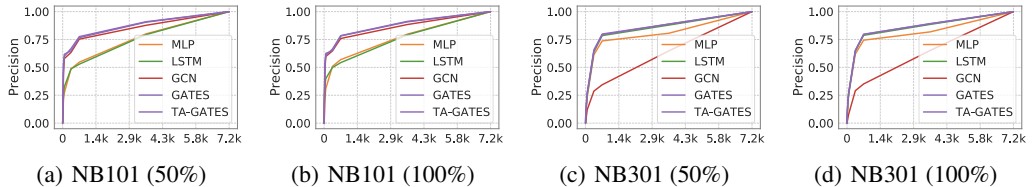

| (a) NB101 (50%) | (b) NB101 (100%) | (c) NB301 (50%) | (d) NB301 (100%) |

Figure A8: Precision@K comparison on the validation split of NB101 and NB301 with sufficient training data. X-axis: K; Y-axis: Precision. The training proportions are 50% and 100%.

Table A6: Comparison of NAS-discovered architectures on ImageNet.

| Method | Top-1 Test Error (%) | #Params (M) |
| --- | --- | --- |
| NASNet-A [57] | 26.0 | 5.3 |
| AmoebaNet-B [31] | 27.2 | 5.3 |
| PNAS [20] | 25.8 | 5.1 |
| DARTS [21] | 26.9 | 4.9 |
| GHN [53] | 27.0 | 6.1 |
| PC-DARTS [48] | 24.2 | 5.3 |
| TE-NAS [2] | 24.5 | 5.4 |
| DI-NAS [28] | 25.3 | 5.2 |
| CATE [49] | 25.0 | 5.8 |
| **TA-GATES** | 24.1 | 5.6 |

so they will not become the bottleneck of NAS efficiency. For example, the hypothetical wall time of the search experiment in Table A5 can be estimated as follows: On NB101, the architecture training time of 79 architectures is about 24 GPU hours; On NB201, the architecture training time of 78 architectures is about 32 GPU hours; On NB301, the architecture training time of 58 architectures is about 90 GPU hours.

## C.3  Ablation Studies

**Ablation study on the iterative encoding process.**  Table A7 shows the influences of the number of time steps $T$. Using 2-step TA-GATES is better on NB101, while on NB201 and NB301, using $T$=3 is better.

About the operation embedding update in Alg. 1 Line 9, we conduct ablation studies on the hyper-parameter $s$, and whether to mask out the updates for nonparametrized operations. As shown in Table A8, the results are not sensitive to the scale $s$. As shown in Table A9, on NB101 and NB201, allowing to update non-parametrized operations is beneficial, while on NB301, masking out the updates of non-parametetrized operations is better.

**Ablation study on the symmetry-breaking technique.**  Table A10 shows an ablation study on the hyper-parameter $\beta$ in Equ. 1. We can see that the results are not sensitive to $\beta$.

Table A7: Kendall's Tau of using different number of time steps $T$.

| Step | NB101 (7290 training) | | | | NB201 (7812 training) | | | | NB301 (5896 training) | | | |
| --- | --- | --- | --- | --- | --- | --- | --- | --- | --- | --- | --- | --- |
| | 1% | 5% | 10% | 50% | 0.5% | 1% | 5% | 10% | 0.5% | 1% | 5% | 10% |
| 2 | 0.6691 | **0.7746** | **0.7821** | **0.8114** | **0.6873** | 0.7640 | 0.8604 | 0.8840 | 0.5726 | 0.6345 | **0.7138** | 0.7302 |
| 3 | **0.6702** | 0.7722 | 0.7772 | 0.8103 | 0.6847 | **0.7708** | **0.8632** | **0.8856** | **0.5755** | **0.6363** | 0.7133 | 0.7313 |
| 4 | 0.6572 | 0.7696 | 0.7753 | 0.8082 | 0.6869 | 0.7699 | 0.8621 | 0.8829 | 0.5728 | 0.6351 | 0.7123 | **0.7331** |

Table A8: Kendall's Tau of using different update scales $s$ of operation embeddings.

| Scale | NB101 (7290 training) | | | | NB201 (7812 training) | | | | NB301 (5896 training) | | | |
|---|---|---|---|---|---|---|---|---|---|---|---|---|
| | 1% | 5% | 10% | 50% | 0.5% | 1% | 5% | 10% | 0.5% | 1% | 5% | 10% |
| 0.01 | 0.6648 | 0.7627 | 0.7774 | 0.8087 | **0.6858** | 0.7641 | **0.8646** | 0.8824 | 0.5711 | 0.6244 | 0.7169 | 0.7323 |
| 0.1 | 0.6686 | **0.7744** | **0.7839** | **0.8133** | 0.6847 | **0.7708** | 0.8632 | 0.8856 | 0.5743 | 0.6267 | 0.7184 | **0.7328** |
| 1.0 | **0.6702** | 0.7722 | 0.7772 | 0.8103 | 0.6772 | 0.7676 | 0.8638 | **0.8871** | **0.5839** | **0.6366** | **0.7189** | 0.7312 |

Table A9: Kendall's Tau of (not) masking out non-parametrized operations.

| Mask | NB101 (7290 training) | | | | NB201 (7812 training) | | | | NB301 (5896 training) | | | |
|---|---|---|---|---|---|---|---|---|---|---|---|---|
| | 1% | 5% | 10% | 50% | 0.5% | 1% | 5% | 10% | 0.5% | 1% | 5% | 10% |
| w. | 0.6654 | 0.7692 | 0.7756 | 0.8050 | **0.6881** | 0.7642 | 0.8590 | 0.8812 | **0.5748** | **0.6351** | 0.7123 | **0.7331** |
| w/o. | **0.6702** | **0.7722** | **0.7772** | **0.8103** | 0.6847 | **0.7708** | **0.8632** | **0.8856** | 0.5743 | 0.6267 | **0.7184** | 0.7328 |

Table A10: Kendall's Tau of using different scale parameters $\beta$ in the "Add" symmetry-breaking technique.

| Encoder | NB101 (7290 training) | | | | NB201 (7812 training) | | | | NB301 (5896 training) | | | |
|---|---|---|---|---|---|---|---|---|---|---|---|---|
| | 1% | 5% | 10% | 50% | 0.5% | 1% | 5% | 10% | 0.5% | 1% | 5% | 10% |
| Add (0.1) | **0.6686** | **0.7744** | **0.7839** | **0.8133** | **0.6753** | 0.7720 | 0.8648 | 0.8879 | **0.5737** | 0.6274 | 0.7079 | 0.7291 |
| Add (0.5) | 0.6677 | 0.7741 | 0.7826 | 0.8120 | 0.6707 | **0.7731** | **0.8660** | **0.8890** | 0.5711 | 0.6335 | 0.7121 | 0.7307 |
| Add (1.0) | 0.6639 | 0.7741 | 0.7822 | **0.8133** | 0.6658 | 0.7649 | 0.8658 | 0.8888 | 0.5728 | **0.6351** | **0.7123** | **0.7331** |

The symmetry-breaking techniques help provide more discriminative architecture encodings. For example, we train a TA-GATES predictor using the "Concat" symmetry-breaking technique on NB201 (100% training data), and the cosine similarity and L2 distance between the predictor's encodings of Arch-A and Arch-B in Fig. 2 (lower) are 0.88 and 1.46. In contrast, a predictor without symmetry breaking outputs exactly the same encodings for Arch-A and Arch-B.

## C.4 Inspection Into Different Time Steps of TA-GATES

We train two four-step TA-GATES on NB201 with ranking loss and regression loss, respectively. Fig. 9(a) shows how the prediction fitness changes over inference time steps. We can see that the prediction fitness gradually improves as the time step increases during inference and surpasses the baseline GATES in step 3 and 4.

We visualize the updates of the contextualized operation embeddings through time steps in Fig. 9(b). Specifically, we collect 163 architecture pairs from the NB201 test split. All the architectures have a Conv3x3 operation from node 1 to node 2, and the architectures in each pair have only one difference: whether there is a skip connection from node 0 to node 2. We use principal component analysis (PCA) to map operation embeddings of the 1-2 Conv3x3 operations to a 2-dim space, and visualize them. We can see that TA-GATES keeps refining the operation embedding across time steps.

## C.5 Inspection Into the Anytime Performance Prediction Task

We show the training curves of TA-GATES and the baseline Multi-GATES in Fig. A10. As shown in Fig. A10(a), we observe a "trade-off" phenomenon of the baseline encoder on NB101, where the KD between the half predicted scores and the half accuracies has an opposite trend with that between the final predicted scores and the final accuracies. We hypothesis that this is because these two supervisory signals contradict each other. In another word, the relative order of half and final accuracies disagree with each other on many pairs of architectures (the KD between the half and final GT accuracy is only 0.4996 on NB101). As a result, directly training baseline encoders with multiple supervisory signals (i.e., the half and final accuracies) even leads to performance degradation compared with Single-GATES.

In contrast, TA-GATES eliminates the "trade-off" phenomenon and actually benefits from multiple supervisory signals (as shown in Table 3, Table A12, Table A13, and Table A14). This is because the

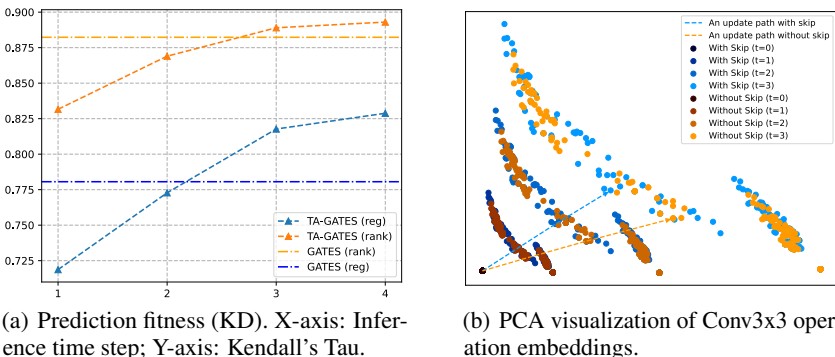

(a) Prediction fitness (KD). X-axis: Inference time step; Y-axis: Kendall's Tau.

(b) PCA visualization of Conv3x3 operation embeddings.

Figure A9: The prediction fitness and operation embeddings at different time steps on NAS-Bench 201. The training proportion is 10%.

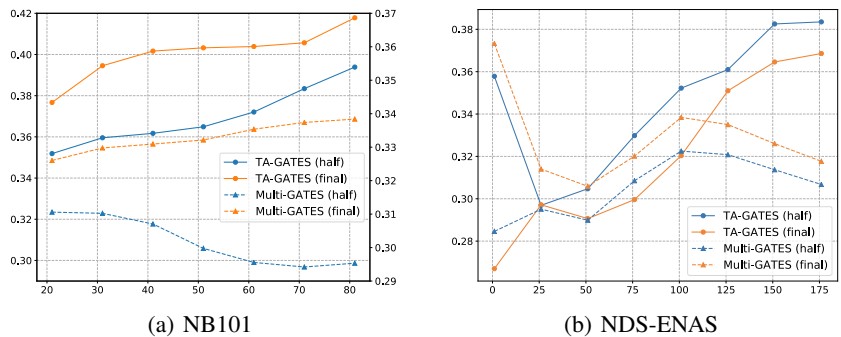

(a) NB101

(b) NDS-ENAS

Figure A10: The trade-off phenomenon on NB101 and NDS-ENAS. X-axis: Training epochs; Y-axis: Kendall's Tau. The training proportion is 0.5% on NB101 and 5% on NDS ENAS.

analogous modeling of the training process in TA-GATES enables it to capture the learning speed of different architectures to some extent.

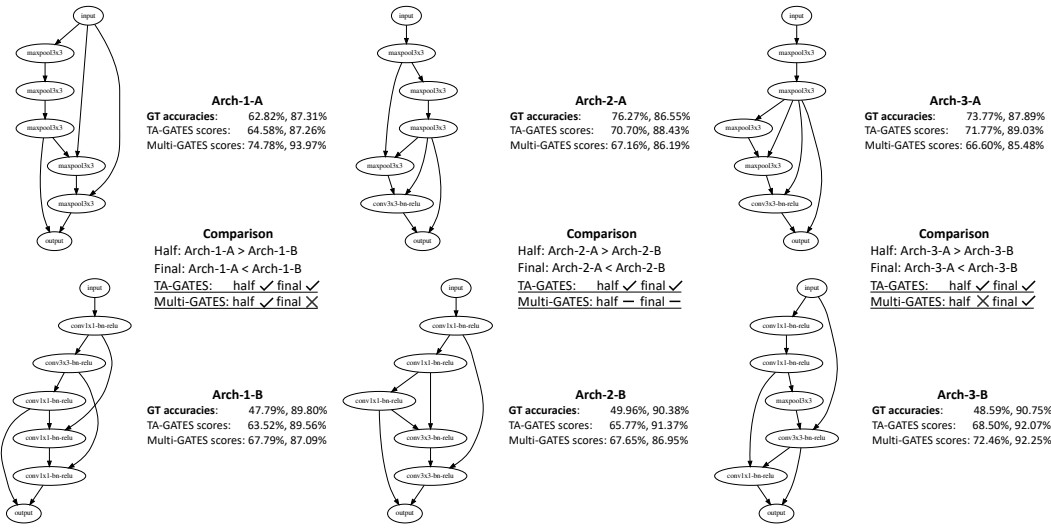

Figure A11: The GT accuracies and predicted scores of some architecture pairs on NB101. TA-GATES can make correct comparisons for the half and final accuracy. Multi-GATES tends to give out the same relative order for the half and final comparisons.

Table A11: Kendall's Tau of anytime training and prediction with sufficient training data. **Upper/Lower**: Kendall's Tau with the half GT accuracy in the middle of training / the final GT accuracy. **Baselines**: "Single-" means to only use one supervisory signal to train a predictor. "Multi-" refers to basic strategy described in Sec. 3.3: The predictor outputs multiple scores and is trained with multiple supervisory signals.

| KD with the half accuracy | | NB101 (7290 training) | | NB301 (5896 training) | |
| --- | --- | --- | --- | --- | --- |
| Encoder | Training | 50% | 100% | 50% | 100% |
| Single-GATES | half | 0.4796 | 0.5385 | 0.7008 | 0.7896 |
| Multi-LSTM | half+final | 0.0518 | 0.0701 | 0.3259 | 0.2797 |
| Multi-GATES | half+final | 0.1413 | -0.0176 | 0.2049 | 0.0637 |
| TA-GATES | half+final | **0.5674** | **0.5538** | **0.7096** | **0.8134** |
| KD with the final accuracy | | NB101 (7290 training) | | NB301 (5896 training) | |
| Encoder | Training | 50% | 100% | 50% | 100% |
| Single-GATES | final | 0.5903 | 0.5875 | 0.6527 | 0.7510 |
| Multi-LSTM | half+final | 0.1473 | 0.1468 | 0.3270 | 0.2674 |
| Multi-GATES | half+final | 0.1818 | -0.0213 | 0.1891 | 0.0784 |
| TA-GATES | half+final | **0.6477** | **0.6909** | **0.7162** | **0.8107** |

To further verify our hypothesis, we show the GT accuracies and predicted scores of some architecture pairs in Fig. A11. Arch-1-A, Arch-2-A, and Arch-3-A with fewer parameters learn at a fast speed and have higher half accuracies, while Arch-1-B, Arch-2-B, Arch-3-B with more parameters learn at a slower speed but have higher final accuracies. We can see that Multi-GATES tends to give out the same relative ranking for the half and final comparisons. Consequently, Multi-GATES gives a wrong final comparison between Arch-1-A and Arch-1-B, and a wrong half comparison between Arch-3-A and Arch-3-B. Multi-GATES also gives out an indistinguishable and wrong comparison between Arch-2-A and Arch-2-B. In contrast, TA-GATES correctly compare the architectures' half and final accuracies in the mean time. This is because the construction of TA-GATES enables it to capture the learning speed characteristics of these architectures to some extent.

As for the ENAS search space, Fig. A10(b) shows that the KDs of baseline predictions with both half and final accuracy drop in the later training period. On the contrary, TA-GATES can obtain constantly increasing KDs during training.

## C.6 Additional Results of Anytime Training And Prediction - Regression Measures and the ENAS Search Space

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

Table A14: Kendall's Tau, Linear Correlation, and Mean Squared Error of anytime training and prediction on NDS ENAS (500 training architectures).

| with the half accuracy | Kendall's Tau | | | | Linear Correlation | | | | Mean Squared Error ($\times 10^{-2}$) | | | |
|---|---|---|---|---|---|---|---|---|---|---|---|---|
| Encoder | 5% | 10% | 50% | 100% | 5% | 10% | 50% | 100% | 5% | 10% | 50% | 100% |
| Multi-GATES | 0.3823 | 0.3841 | 0.3554 | 0.3625 | 0.1280 | 0.1300 | 0.1100 | 0.0954 | **0.8581** | 2.0143 | 1.0979 | 0.9208 |
| TA-GATES | **0.4257** | **0.4391** | **0.4592** | **0.4755** | **0.1679** | **0.1740** | **0.3426** | **0.4229** | 0.8748 | **1.1779** | **0.9933** | **0.8123** |

| with the final accuracy | Kendall's Tau | | | | Linear Correlation | | | | Mean Squared Error ($\times 10^{-2}$) | | | |
|---|---|---|---|---|---|---|---|---|---|---|---|---|
| Encoder | 5% | 10% | 50% | 100% | 5% | 10% | 50% | 100% | 5% | 10% | 50% | 100% |
| Multi-GATES | 0.3823 | 0.3841 | 0.3554 | 0.3625 | 0.0773 | 0.0779 | 0.0638 | 0.0502 | 0.9349 | 2.3654 | **1.0625** | 0.9684 |
| TA-GATES | **0.4257** | **0.4391** | **0.4592** | **0.4755** | **0.1027** | **0.1118** | **0.2974** | **0.3754** | **0.9310** | **1.2173** | 1.0934 | **0.9422** |