# OpenReview forum: "TA-GATES: An Encoding Scheme for Neural Network Architectures"
_NeurIPS.cc/2022/Conference — NeurIPS 2022 Accept_

### Official Review · Reviewer_zsxx · 2022-07-11

**Rating:** 7
**Confidence:** 4
**Soundness:** 3 good
**Presentation:** 3 good
**Contribution:** 3 good

**Summary:**

The authors present a novel encoding scheme for neural network architectures based on the GATES method. They propose a significant change to the GATES approach that trains a different embedding for each of the operations even if they are of the same type. To achieve this they present a series of methods for symmetry breaking that allow the embeddings for each operation to vary independently. They perform an exhaustive series of experiments verifying and ablating their claims to show that their approach TA-GATES is indeed superior to the alternative encoding approaches they consider.

**Questions:**

See the Weaknesses part of the Strengths and Weaknesses section.

**Limitations:**

The authors have adequately addressed the limitations of their work.

**Strengths And Weaknesses:**

# Strengths
- This is a well-written paper. The prose is clear and the arguments are comprehensible. Enough detail is provided to understand the approach suggested and to understand the experimental setup.
- The extensive experiments seem to indicate that indeed TA-GATES is a superior embedding to the suggested alternatives.
- The area explored by the paper is extremely relevant as having better encodings for architectures can lead to better downstream models and better analysis of architectures based on the encodings.
- The authors show the added benefits of TA-GATES for anytime performance evaluation that can be extremely useful when partial learning curve information is required.

# Weaknesses
- They do not reference other NN encoding schemes, like those suggested in [1].
- Their argument that the architecture should include variation based on the specific parameters of the model seems suspect. While clearly using different embeddings for the same operations does seem to work well, the argument that the architecture should incorporate this seems incorrect since most of the time the specific parameters of the operation do not seem to be relevant when designing architectures.
- The authors don't include the runtime of TA-GATES. While I assume that the runtime is comparable to GATES it would help to know the specific run times of their approach and compare it to the other approaches

# References
[1] White, C., Neiswanger, W., Nolen, S., & Savani, Y. (2020). A study on encodings for neural architecture search. Advances in Neural Information Processing Systems, 33, 20309-20319.

---

> ### Author Response · Authors · 2022-08-02
> **Reply to Reviewer zsxx**
>
> Thank you for the recognition of our work's relevant significance, presentation, extensive experiments, and the application potential of the improved anytime performance prediction ability, as well as the constructive questions. We answer the questions below.
>
> **Q1** Should reference other NN encoding schemes, like those suggested in [1].
>
> **A1** Thanks for the suggestion. We'll reference and discuss more papers provided by [1]. And we also conduct some more baseline comparisons as suggested. Please refer to the "Add Baseline Encoder Comparison" section in the common comment.
>
> **Q2** While clearly using different embeddings for the same operations does seem to work well, the argument that the architecture should incorporate this seems incorrect since most of the time the specific parameters of the operation do not seem to be relevant when designing architectures.
>
> **A2**: Thanks for the question, it's really worth discussing, and we'd like to share our opinion. It's not that we consider such detailed things like "what the specific parameters of an operation are". Instead, we only need the encoder to have the ability to give different embeddings for different operations (even those of the same type), or borrowing the word from CATE, "contextualized embeddings". In another word, operations with "different functionalities" should be discriminable when modeling the architecture. The "functionality" is a somewhat abstract notion, not some specific parameters.
>
> Actually, during manual architecture design, "even if two operations are of the same type, they have different functionalities according to their architectural context" might be a self-evident prior knowledge. For example, we people would intuitively know that the two Conv3x3 in Fig.1 play a different role.
>
> Now that what is TA-GATES and how does TA-GATES realizes the above-mentioned discriminative power: TA-GATES is an architecture modeling method that could be used in the design (especially automatical design) of architectures. It enables the above-mentioned discriminative modeling of operations in a natural way. Specifically, the core design intuition behind TA-GATES is that an architecture not only describes what the forward computation semantics are, but also determines the weight training dynamics. Therefore, we design an ``encoding by training-mimicking'' way to encode the architecture by refining the embedding of each operator. In this way, every operation could get different "contextualized" embedding during the encoding process according to its architectural context (what the overall architecture is and where the operation is). Extensive experiments verify that TA-GATES could give better predictions of architecture performances and also capture training dynamics during architecture encoding.
>
> **Q3** The authors don't include the runtime of TA-GATES. While I assume that the runtime is comparable to GATES it would help to know the specific run times of their approach and compare it to the other approaches.
>
> **A3** Thanks and we'll add these information. Currently, we discuss the training and inference time of TA-GATES in Section C.1. We run predictor training and inference using _NVIDIA RTX 3090 GPU_ and _AMD EPYC 7H12_. We also estimate the hypothetical wall time of the search experiment in Sec. C.1 as follows: On NB101, the architecture training time of 79 archs is about 24 GPU hours; On NB201, the architecture training time of 78 archs is about 32 GPU hours; On NB301, the architecture training time of 58 archs is about 90 GPU hours. While the training time of GATES & TA-GATES is around 1.5min and 2min, so this would definitely not become the bottleneck of NAS efficiency.

---

> > ### Comment · Reviewer_zsxx · 2022-08-09
> > **Thank you for your response**
> >
> > Thank you for the additional metrics provided and for the comments addressing my concerns. It seems like TA-GATES is an effective encoding approach compared to several standard baselines across NAS tasks. I will increase my score.

---

> > > ### Author Response · Authors · 2022-08-10
> > > **Thanks for your response!**
> > >
> > > We're encouraged to see your recognition to our work! Also, your constructive feedback really helps us to improve our paper. We'll follow the suggestions to prepare the revision.

---

### Official Review · Reviewer_64UJ · 2022-07-12

**Rating:** 6
**Confidence:** 4
**Soundness:** 3 good
**Presentation:** 3 good
**Contribution:** 3 good

**Summary:**

This paper recommends a new GCN encoder based performance predictor built on top of the GATES encoder/predictor. While most encoders assign the same embedding for a given operation irrespective of their position in the architecture, TA-GATES proposes a predictor that captures the training process of the architecture and assigns different embeddings for a given operation.

Given an adjacency matrix representation of the architecture $a$ and the input information E, for each time step, it computes the information flow of $a$ through a GCN similar to the GATES paper. This simulates the forward propagation of NN training. Then this information is transformed by passing through an MLP and  the resulting output along with $a^{T}$ is in turn passed through the GCN with different weight parameters for information propagation. To simulate backpropagation of the training, they compute derivates and use chaining rule in their second information propagation. The output of the backward propagation from the final timestep is the architecture embedding. Similar to GATES, they have a pairwise ranking loss and MSE loss. But rather than predicting the final accuracy, TA-GATES predicts the accuracy for every half epoch and full epoch time steps. This is seamlessly learnt because TA-GATES is trained for every time-step.

Further, to avoid the operations from having the same embeddings initially, they use some symmetric breaking techniques such as (1) adding random noise (2) add an vector comprised of zero-cost metrics such as grad_norm, snip etc. only initially or at every time step



**Questions:**

1) For table 1 and Table 3, Please compare against more recent performance predictors for some search spaces at least. [1] has open sourced several performance predictors. XGBoost seems to be competitive.

2) For Table1, please include the kendall Tau metrics for 50% and 100% for NB201 and 100% for NB301.

3) Similarly, please include Precision @ K for all the predictors when predicted on 50% and 100% of the training data.

4) For Table3 also, it would be good to know how Multi-Gates fares against TA-Gates when trained on 100% of the training data.

5) For Table A5, please list the accuracies of the architectures found when the predictors are trained on 100% of the training data.

[1]How Powerful are Performance Predictors in Neural Architecture Search?, White et al.

**Limitations:**

See the weakness and questions section.

While the idea of training the surrogate model that captures the training dynamics of the model is an interesting idea, it would be good to understand if it is really essential. Early stopping methods, some of which are listed in [1] are able to predict the accuracies at a given epoch by just extrapolating the learning curves. For your accuracy prediction at half of the training epochs, it would be compare against those for Table 3.

**Strengths And Weaknesses:**

Strengths:
TA-GATES is able to predict the accuracy for every epoch and not just the final accuracy. So in addition to using it as standalone for NAS, this would be useful to select models that perform well by training for fewer epochs where we have limited training budget.
TA-GATES is able to out perform all the other baselines when trained on limited data.

Weakness:
Despite their new training strategy and improved correlation metrics, TA-GATES is not able to rate the top-k architectures significantly better than GATES as indicated by Precision @ K.
The range of accuracies of architectures found in NDS-Darts is a lot lesser than NAS-Bench 201. So in general, it would be good to demonstrate that the surrogate models work effectively there. NAO was also initially developed on the DARTS search space.

---

> ### Author Response · Authors · 2022-08-02
> **Reply to Reviewer 64UJ (1/2)**
>
> Thank you for the recognition of our work's application potential and superior performances, as well as the constructive questions. We answer the questions below.
>
> **Q1** For Table 1 and 3, Please compare against more recent performance predictors for some search spaces at least. [1] has open sourced several performance predictors. XGBoost seems to be competitive.
>
> **A1**  Thanks for the suggestion. We add discussions or comparisons to several baselines as follows. Please refer to the "Add Baseline Encoder Comparison" section in the common comment.
>
> We also add XGBoost baseline in Table 3 as follows:
>
> | KD with the half accuracy (NB101)  |   1%    |  5%  |10%|50%|100%|
> |---|---|---|---|---|---|
> | Single-XGBoost       | 0.2830 |0.3476 | 0.3430 |0.3820|0.3971|
> |TA-GATES            |  **0.3921**|**0.4615**  | **0.4805**|**0.5674**|**0.5538**|
>
> | KD with the final accuracy (NB101)  |   1%    |  5%  |10%|50%|100%|
> |---|---|---|---|---|---|
> | Single-XGBoost       | 0.4520 |**0.5987** | 0.5680 | 0.5677|0.6175|
> |TA-GATES            | **0.5463**|0.5850  | **0.5950**|**0.6477**|**0.6909**|
>
>
> **Q2** For Table1, please include the kendall Tau metrics for 50% and 100% for NB201 and 100% for NB301.
>
> **A2** The results are as follows. TA-GATES still outperforms the baselines. One exception is that on NB201, using 50% (3906) or 100% (7813) architectures as the training data, GATES and TA-GATES have similar saturating performances as the search space is small (15k architectures in total, 6k deiso), and architectures are easy to distinguish. Usually, one does not want to train this many architectures for modeling a search space like NB201, and we think training 10% * 7813 = 781 architectures is already considered expensive for NB201. We'll add these results and discussions.
>
> | Encoder              |    NB201 (50%)    |    NB201 (100%)   |    NB301 (100%)    |
> |---|---|---|---|
> | MLP                 | 0.8205 (0.0050) | 0.8733 (0.0011) | 0.6501 (0.0014) |
> | LSTM                | 0.8757 (0.0018) | 0.9008 (0.0013) |0.7672 (0.0009) |
> | GCN             | 0.7733 (0.0000) | 0.8257 (0.0000) | 0.3256 (0.0016) |
> |GATES            | 0.9155 (0.0090) | **0.9259 (0.0013)** | 0.7670 (0.0053) |
> |TA-GATES         |  **0.9181 (0.0041)**|0.9228 (0.0041)  |  **0.7766 (0.0033)**|
>
>
> **Q3** Similarly, please include Precision @ K for all the predictors when predicted on 50% and 100% of the training data.
>
> **A3** The results are as follows. We can see that except for only a few points where GATES is marginally better, TA-GATES achieves a consistently better P@TopK than all baselines.
>
> | NB101 (50%)              |    P@Top 0.1% |P@Top 0.5% |P@Top 1%|P@Top 5%|P@Top 10%|P@Top 50%|P@Top 100%|
> |----------------------|--------------|--------------|--------------|--------------|--------------|--------------|--------------|
> | MLP                 | 0.0158|0.2191|0.2716|0.4859|0.5468|0.8003|1|
> | LSTM                | 0.0793|0.2469|0.3256|0.4862|0.5287|0.7933|1 |
> | GCN             | 0.0793|0.4475|0.5787|0.6339|0.7532|0.8770|1 |
> |GATES            | 0.0634|0.5246|**0.6172**|0.6483|0.7668|0.9051|1 |
> |TA-GATES         |**0.1746**|**0.5308**|0.6003|**0.6681**|**0.7759**|**0.9091**|**1**|
>
> | NB101 (100%)              |    P@Top 0.1% |P@Top 0.5% |P@Top 1%|P@Top 5%|P@Top 10%|P@Top 50%|P@Top 100%|
> |----------------------|--------------|--------------|--------------|--------------|--------------|--------------|--------------|
> | MLP                 |0.0317|0.2191|0.3086|0.5091|0.5697|0.7977|1|
> | LSTM                | 0.1746|0.3333|0.3996|0.4984|0.5426|0.7920|1|
> | GCN             |0.0158|0.4722|0.5972|0.6473|0.7591|0.8841|1 |
> |GATES            | 0.1428|0.5123|**0.6250**|0.6504|0.7817|0.9080|1 |
> |TA-GATES         |**0.2380**|**0.5339**|0.6064|**0.6642**|**0.7847**|**0.9125**|**1**|
>
> | NB301 (50%)              |    P@Top 0.1% |P@Top 0.5% |P@Top 1%|P@Top 5%|P@Top 10%|P@Top 50%|P@Top 100%|
> |----------------------|--------------|--------------|--------------|--------------|--------------|--------------|--------------|
> | MLP                 | 0.0588|0.1418|0.2372|0.6126|0.7375|0.8059|1|
> | LSTM                | 0.0490|0.1673|0.2526|0.6399|0.7850|0.8790|1 |
> | GCN             |0.0098|0.0601|0.1202|0.2872|0.3439|0.6476|1|
> |GATES            |0.0740|0.1599|0.2712|0.6421|0.7929|**0.8881**|1 |
> |TA-GATES         |**0.0762**|**0.1716**|**0.2716**|**0.6570**|**0.7990**|0.8880|**1**|
>
> | NB301 (100%)              |    P@Top 0.1% |P@Top 0.5% |P@Top 1%|P@Top 5%|P@Top 10%|P@Top 50%|P@Top 100%|
> |----------------------|--------------|--------------|--------------|--------------|--------------|--------------|--------------|
> | MLP                 | 0.0620|0.1647|0.2555|0.6192|0.7440|0.8183|1|
> | LSTM                | 0.0555|0.1732|0.2604|0.6455|0.7872|0.8853|1 |
> | GCN             |0.0065|0.0411|0.0800|0.2908|0.3514|0.6519|1|
> |GATES            |0.0631|**0.1925**|**0.2738**|0.6466|0.7957|0.8905|1 |
> |TA-GATES         |**0.0697**|0.1790|0.2694|**0.6536**|**0.7971**|**0.8923**|**1**|

---

> > ### Comment · Reviewer_64UJ · 2022-08-07
> > **Thank you for your response**
> >
> > Thanks for adding all the metrics that were requested for. TA-GATES seems to be performing better than all the others on most datasets and with lesser data. However, GATES does seem to be a strong competitor in some cases, esp in NB301.  I will increase my score.

---

> > > ### Author Response · Authors · 2022-08-08
> > > **Thanks for your response!**
> > >
> > > We're encouraged to see that our responses address your major concern. Again, thank you for your constructive suggestions and insightful questions, they really help us to improve our paper.

---

> ### Author Response · Authors · 2022-08-02
> **Reply to Reviewer 64UJ (2/2)**
>
> **Q4** For Table3, it would be good to know how Multi-Gates fares against TA-Gates when trained on 100% of the training data.
>
> **A4** The results are as follows.
>
> | KD with the half accuracy  |    NB101 (100%)    |    NB301 (50%)   |    NB301 (100%)    |
> |----------------------|--------------|--------------|--------------|
> | Single-GATES       | 0.5385 |0.7008 | 0.7896 |
> | Multi-LSTM         | 0.0701 | 0.3259 |0.2797 |
> | Multi-GATES        | -0.0176 | 0.2049 |0.0637 |
> |TA-GATES            |  **0.5538**|**0.7096**  | **0.8134**|
>
> | KD with the final accuracy  |    NB101 (100%)    |    NB301 (50%)   |    NB301 (100%)    |
> |----------------------|--------------|--------------|--------------|
> | Single-GATES       | 0.5875 |0.6527 | 0.7510 |
> | Multi-LSTM         | 0.1468 | 0.3270 |0.2674 |
> | Multi-GATES        | -0.0213 | 0.1891 |0.0784 |
> |TA-GATES            | **0.6909**|**0.7162**  | **0.8107**|
>
>
> **Q5** For Table A5, please list the accuracies of the architectures found when the predictors are trained on 100% of the training data.
>
> **A5** The results are as follows.
>
> | Encoder              |    NB101 (7920)    |    NB301 (5896)   |
> |----------------------|--------------|--------------|
> | MLP                 | 0.9440 (0.0022) | 0.9447 (0.0017) |
> | LSTM                | 0.9434 (0.0027) | 0.9442 (0.0015) |
> | GCN             | 0.9434 (0.0025) | 0.9378 (0.0056) |
> |GATES            | 0.9428 (0.0017) | 0.9448 (0.0017) |
> |TA-GATES         |  **0.9443 (0.0024)**|**0.9452 (0.0018)**  |
>
>
> **Q6** The range of accuracies of architectures found in NDS-Darts is a lot lesser than NAS-Bench 201. So in general, it would be good to demonstrate that the surrogate models work effectively there.
>
> **A6** Yes, we agree that NB201 is easy as the search space is small and the architectures are easy to distinguish. And we have already conducted experiments on NB301 (the DARTS search space) and also NDS-ENAS (also an operation-on-edge search space, the key differences with DARTS are: the number of nodes, operation types, and each architecture only concatenates loose-end intermediate nodes as the output), in which the architecture accuracy distribution is far more concentrated than that on NB201. These two benchmarks, especially NB301, have similar properties with NDS-DARTS, and TA-GATES presents better performances consistently on them.
>
> **Q7** It would be good to understand if it is really essential. Early stopping methods, some of which are listed in [1] are able to predict the accuracies at a given epoch by just extrapolating the learning curves.
>
> **A7** Thanks for this constructive question. Actually, we have the plan of developing learning curve extrapolation tool based on TA-GATES in future work. We'll add discussions on the learning curve extrapolation methods (e.g., Speeding up automatic hyperparameter optimization of deep neural networks by extrapolation of learning curves) and more future work discussions. These methods assume the training curve could be described by a set of parametric function families, and for each training process of a new configuration (e.g., a new architecture), they fit the parameters of the function families using the true valid performances of the early training curve, and then extrapolate to get the performance estimations at further epochs.
>
> TA-GATES has the potential to be used as a black-box and learnable extrapolation tool, and this is certainly the most important future exploration of this work. We can share our idea here: In the exploration process, we have a pretrained TA-GATES predictor. Given an architecture $a$, we train it for several epochs and evaluate its true valid performances e.g., at epoch 10,20,30. Then, we conduct a few-step finetune of the TA-GATES using the true valid performances of $a$ at epoch 10,20,30 as the supervisory signals for the corresponding time steps (e.g., t=1,2,3). Finally, to get the extrapolated predictions (e.g., at epoch 40,50,60), we just get the prediction of TA-GATES at these corresponding time steps (e.g., t=4,5,6). Note that there is a few-step adaption/fitting process of TA-GATES to the early training curve of each architecture. As for the training process of the TA-GATES predictor, we think if the vanilla training of TA-GATES does not work well, we could use a MAML-like episode training method.
>
> Note that the essential difference is that TA-GATES takes the architecture description as input, i.e., architecture-dependent. After the meta-training process of TA-GATES, it models how the architecture determines its training curve. That is to say, during the extrapolation, instead of solely relying on the early training curve points, a learnable architecture-dependent extrapolation (with per-architecture few-step adaption) method can essentially utilize the similarities between architectures and the knowledge of learning curves of other architectures, and has the potential to make better extrapolations.

---

### Official Review · Reviewer_dXbo · 2022-07-13

**Rating:** 4
**Confidence:** 4
**Soundness:** 3 good
**Presentation:** 3 good
**Contribution:** 3 good

**Summary:**

The authors consider encoding schemes for neural architecture search (NAS). While current encodings model the data flow through the architecture operations by encoding  each operation, the current authors propose a new encoding based on the idea that different operations need different encodings, since each operation has its own separate weights that are learned throughout architecture training. Specifically, they perform multiple forward and backward passes on the architecture and output the trained embeddings as the encoding. They also give symmetry-breaking techniques (similar idea as random initialization). They test their technique on NAS-Bench-101, 201, 301, and ENAS.

**Questions:**

Can the authors show the usefulness of TA-GATES in NAS by running full NAS experiments? Can the authors compare to more encoding-based methods such as the examples above?

Can the authors include runtimes, CPU/GPU information, limitations/societal impact?

**Limitations:**

The authors did not discuss societal impact of their work, and I also did not see a discussion on limitations. Remember that the NeurIPS page limit was increased from 8 to 9 pages to give space for the questions in the author checklist, so these sections should be taken more seriously.

**Strengths And Weaknesses:**

### Strengths

Intuitive and well-performing. The TA-GATES intuition, i.e., operations should not be encoded the same way because they have trainable parameters, makes sense, and TA-GATES is an intuitive solution to this. The symmetry-breaking technique also makes sense, and overall these techniques are shown to achieve strong performance on four benchmarks.

They released code with a detailed README, so reproducibility is high.

Good ablations: ablation for number of time steps of TA-GATES, and ablation for symmetry-breaking technique.

The anytime performance prediction is a very interesting application of TA-GATES.

### Weaknesses

No NAS experiments. All of the experiments compute rank correlation or precision for performance prediction. To show the usefulness of TA-GATES in NAS, the authors should run full NAS experiments.

Similarly, the method is only compared to four other performance prediction models: MLP, LSTM, GCN, and GATES. But, the authors could include even more. For example, there are other encoding-based works such as
“NASBOWL”, “Contrastive Embeddings for Neural Architectures”, “Does Unsupervised Architecture Representation Learning Help Neural Architecture Search?”, “CATE”, and. Some of these works learn architecture embeddings, which is similar in spirit to TA-GATES, so it is important to discuss and compare to them.

The specific TA-GATES approach seems to be specific for each search space (based on all of the details in Section B.3) so it would need to be tuned for each new search space.

Repeatedly refer to GATES and D-VAE as state-of-the-art techniques. But these have been out for more than two years. There are other methods now, such as the ones listed in the second point above.

The runtime of their method should be included in the main body, as well as total CPU and/or GPU usage.

No discussion on limitations or societal impact (see below).

---

> ### Author Response · Authors · 2022-08-02
> **Reply to Reviewer dXbo**
>
> Thank you for the recognition of our work's working intuition, reproducibility, ablation, and interesting application, as well as the constructive questions. We answer the questions below.
>
> **Q1** All of the experiments compute rank correlation or precision for performance prediction. The authors should run full NAS experiments.
>
> **A1** Currently, we put some NAS experiments in Appendix C.1. According to the suggestion, we add some search experiments on NB101 & NB301 in the following table, where the number in the parenthesis in the header is the training architecture number of the predictor (and the inner search subroutine is a random-search subroutine described in Appendix C.1). We also train our found architecture on DARTS on CIFAR-10, and the architecture got a test error of 2.46% with 4.1M parameters.
>
> | Encoder  |    NB101 (39)    |    NB101 (396)   |    NB301 (5)    | NB301 (2948)  | NB301 (5896)   |
> |---|---|---|---|---|---|
> | MLP | 0.9280 (0.0142) |0.9402 (0.0027) |0.9374 (0.0071)  | 0.9448 (0.0017) |0.9447 (0.0017) |
> | LSTM | 0.9359 (0.0067)  |0.9416 (0.0024) |0.9380 (0.0034) |0.9444 (0.0019) |0.9442 (0.0015)  |
> | GCN |0.8750 (0.1147) | 0.9417 (0.0025) | 0.9254 (0.0127) |0.9384 (0.0053) | 0.9378 (0.0056) |
> |GATES | 0.9407 (0.0027) | 0.9417 (0.0014) | 0.9398 (0.0045) |0.9450 (0.0014) | 0.9448 (0.0017) |
> |TA-GATES |  **0.9424 (0.0024)** | **0.9429 (0.0020)**  |  **0.9417 (0.0045)** |**0.9450 (0.0016)**  | **0.9452 (0.0018)**  |
>
> We focus on the direct fitness measures of the predictor in the main text. Using these measures gets rid of other confounding factors in the NAS process, and most directly reveals the quality of modeling methods. Measures like P@Ks have a direct correspondence to the NAS process (some inner-search subroutines would select topK architectures according to the predicted scores).
>
> **Q2** Repeatedly refer to GATES/D-VAE as SOTA techniques. There are other methods, can the authors compare more encoding-based methods?
>
> **A2** Thanks for this constructive suggestion and pointing out related work. We add discussions or comparisons to several baselines as follows. Please refer to the "Add Baseline Encoder Comparison" section in the common comment.
>
> **Q3** The specific TA-GATES approach seems to be specific for each search space (based Sec. B.3) so it would need to be tuned for each search space.
>
> **A3** Actually, many of the details in Sec. B.3 are about the construction of our baseline encoders. As for TA-GATES, we do not conduct any hyperparameter search or manual adjustment for the TA-GATES configurations in different search spaces. Only the number of TA-GATES layers is set to match the longest path in the search space, and for other configurations (e.g., number of hidden units, embedding sizes), we just set them as the same as the baseline GATES. Another thing we'd like to mention is that the design in Sec. A.2 to handle different cases is also general instead of a per-search-space design. For example, Equ. A4 shows the only modification needed to handle an OOE architecture (the main text shows the corresponding equation when handling OON architectures). And the method of handling multiple-cell and multiple-input-node architectures is general across search spaces.
>
> **Q4** The runtime should be included in the main body.
>
> **A4** Thanks and we'll add these information into the main text. Currently, we discuss the training and inference time of TA-GATES in Section C.1. We run predictor training and inference using _NVIDIA RTX 3090 GPU_ and _AMD EPYC 7H12_. We also estimate the hypothetical wall time of the search experiment in Sec. C.1 as follows: On NB101, the architecture training time of 79 archs is about 24 GPU hours; On NB201, the architecture training time of 78 archs is about 32 GPU hours; On NB301, the architecture training time of 58 archs is about 90 GPU hours. While the training time of GATES & TA-GATES is around 1.5min and 2min, so this would definitely not become the bottleneck of NAS efficiency.
>
> **Q5**  Include limitations/societal impact.
>
> **A5** Thanks and we'll add more discussions. We do not think TA-GATES has negative societal impacts, and Sec. 6 discusses the application potential and impacts of TA-GATES for general AutoML. We attribute the potential impact of TA-GATES to its unified methodology of ``encoding by mimicking''. Rather than crafting separate encoder designs for different types of factors in the NN training pipeline, TA-GATES could naturally and jointly encode these factors. A notable limitation is that, like other training-driven predictors, TA-GATES could not conduct cross-search-space comparisons. We think this is a very worth-studying future topic to bridge the gap between **zero-cost predictors** (could conduct cross-space comparison but has non-satisfying predictions) and **training-driven & space-representation-related predictors** (have stronger predictive power and could benefit from training data, but cannot conduct cross-space comparison).

---

> > ### Comment · Reviewer_dXbo · 2022-08-08
> > **Thank you for your response**
> >
> > I like the experiments with NASBOWL, SemiNAS, XGBoost, and CATE, and the discussion on the similarities and differences of these methods and of Arch2Vec and Contrastive Embeddings. The new experiments look good.
> >
> > **Q1**
> > I now see that I missed the NAS experiments in the appendix. Although, it would be more realistic to run TA-GATES with a higher performing search algorithm than just random search. For example, any guided search algorithm.
> >
> > **Q2 - Q5**
> > Thank you for addressing my concerns.
> >
> > The new comparisons to the other encoders look good. However, the empirical conclusions would be stronger if there were better NAS experiments -- this would prove the usefulness of TA-GATES in NAS. I understand that the authors did not want a more complex NAS method to not have more confounding variables. But, two experiments that seem interesting would be (1) fixing a NAS framework, and comparing TA-GATES + that framework to other encodings with the same framework. (2) Comparing TA-GATES + a NAS algorithm to other end-to-end NAS algorithms.

---

> > > ### Author Response · Authors · 2022-08-09
> > > **Further responses**
> > >
> > > Thanks for your response and the recognition of our new experiments and the discussion! Based on your suggestion, we combine TA-GATES with Tournament Evolutionary Search to conduct an NAS experiment on NB201, and compare the result with other end-to-end baselines in the following table. The baseline results except Evo+GATES are directly taken from the ReNAS paper. Due to the limited time, we only provide the new results with tournament-based evolutionary search for now, and we'll add more search frameworks with more configurations to this table in later revision.
> > >
> > > | Method| test acc. on CIFAR-10 |
> > > |----|----|
> > > | NPENAS | 91.52 |
> > > | REA | 93.92 |
> > > | NASBOT | 93.64 |
> > > | REINFORCE | 93.85 |
> > > | BOHB | 93.61 |
> > > | ReNAS | 93.99 |
> > > | Evo + GATES | 93.92  |
> > > | **Evo + TA-GATES** |  93.995 |
> > >
> > > The concrete search configurations are: we use 5 different initial populations, 9 different seeds in training predictor. We use 5% training data for training the predictor, the evolutionary population size is 20, the tournament size is 10, and we run 50 evolutionary steps.
> > >
> > > Please let us know if there are any more unsolved questions we can explain.

---

### Author Response · Authors · 2022-08-02
**Reply to a common question of all reviewers (1/2)**

We thank all reviewers for their constructive feedback. The questions and suggestions are very helpful for us to improve our work. Here we post a reply to a common question of all reviewers "should reference, discuss, and compare more baselines".

# Add Baseline Encoder Comparison

1. NASBOWL[2] proposes to use the WL graph kernel with multiple iterations in the Gaussian Process surrogate. We run their code using our evaluation setting (data split, shuffle seeds, training seeds).
2. CATE[3] proposes a self-supervised training method to pretrain a transformer-based encoder, and use the pretrained embedding to predict architecture accuracies. We use their pretrained encoding on NB101 and find that CATE's fitness is not satisfying: the maximum Kendall's Tau is lower than 0.3 even if we use all the training data. We've confirmed with the author by email that (1) our usage of their code and package versions are all correct; (2) they did not run the ranking correlation comparison, and maybe their Kendall's Tau is not high. Therefore, we compare with their final search results. After carefully studying CATE, we identify an interesting link between CATE and our work: The motivation behind CATE's transformer design is to capture "deep contextualized information" of operations. They use reachability-masked attention to aggregate operation embeddings for several iterations. In this way, CATE can get different operation embeddings for different operations. In contrast, TA-GATES proposes a more natural way to get "contextualized embeddings" for operations and enable more potentials such as anytime prediction for multi-fidelity AutoML. We'll add this discussion to the related work.
3. Arch2Vec[4] explores training architecture representations in an unsupervised way. And it uses a GIN encoder. To compare GIN with TA-GATES, we train their GIN using the same supervised training setting as our other experiments.
4. XGBoost in "How Powerful are Performance Predictors in Neural Architecture Search?"[1]. We use the code in NASLib[6].
5. SemiNAS[5] uses an LSTM encoder. As our work focuses on the encoder design and TA-GATES should be orthogonal with the training method, we follow their encoder construction setting and train with our supervised training setting.
6. We thank the reviewer for pointing out "Contrastive Embeddings for Neural Architectures"[7], it's an interesting work whose training strategy and usage of zero-shot information can be combined with TA-GATES to enable universal across-space comparison (as we also discussed in A5 in the reply to Reviewer dXbo).

# References

[1] White, Colin, et al. "How powerful are performance predictors in neural architecture search?." Advances in Neural Information Processing Systems 34 (2021): 28454-28469.

[2] Ru, Binxin, et al. "Interpretable Neural Architecture Search via Bayesian Optimisation with Weisfeiler-Lehman Kernels." International Conference on Learning Representations. 2020.

[3] Yan, Shen, et al. "Cate: Computation-aware neural architecture encoding with transformers." International Conference on Machine Learning. PMLR, 2021.

[4] Yan, Shen, et al. "Does unsupervised architecture representation learning help neural architecture search?." Advances in Neural Information Processing Systems 33 (2020): 12486-12498.

[5] Luo, Renqian, et al. "Semi-supervised neural architecture search." Advances in Neural Information Processing Systems 33 (2020): 10547-10557.

[6] Ruchte, Michael, et al. "NASLib: a modular and flexible neural architecture search library." (2020).

[7] Hesslow, Daniel, and Iacopo Poli. "Contrastive embeddings for neural architectures." arXiv preprint arXiv:2102.04208 (2021).

---

> ### Author Response · Authors · 2022-08-02
> **Reply to a common question of all reviewers (2/2)**
>
> | KD on NB101       |    1%    |    5%  |    10%    |    50%   |  100%|
> |----------------------|--------------|--------------|--------------|--------------|--------------|
> | NASBOWL                 | 0.5850 (0.0232) | 0.6416 (0.0241) | 0.6536 (0.0193) | 0.6833 (0.0022) | 0.6872 (0.0000) |
> | SemiNAS               | 0.5273 (0.0589)|0.6055 (0.0294)|0.5953 (0.0279)|0.6040 (0.0284)|0.6043 (0.0179) |
> | XGBoost             | 0.4517 (0.0470) | 0.5987 (0.0365) | 0.5680 (0.0125) | 0.5677 (0.0077) | 0.6175 (0.0000) |
> |TA-GATES         |  **0.6686 (0.0338)**|**0.7744 (0.0211)**  |  **0.7839 (0.0063)**|**0.8133 (0.0053)**  | **0.8217 (0.0057)** |
>
> | KD on NB201 | 0.1%| 0.5% | 1%    |    5%  |    10%    |    50%   |  100%|
> |----------|----------|----------|--------------|--------------|--------------|--------------|--------------|
> | NASBOWL  | 0.4980 (0.0408) |  0.6674 (0.0077) | 0.5912 (0.0874) | 0.7259 (0.0098) | 0.7625 (0.0083) | 0.8080 (0.0022) | 0.8077 (0.0033) |
> | XGBoost  |0.0706 (0.1238) | 0.3719 (0.0560)      | 0.4178 (0.0288) | 0.6412 (0.0053) | 0.7084 (0.0123) | 0.7977 (0.0048) | 0.8312 (0.0000) |
> |TA-GATES  |**0.5382 (0.0478)** |**0.6707 (0.0256)**   |  **0.7731 (0.0249)**|**0.8660 (0.0060)**  |  **0.8890 (0.0049)**|**.9181 (0.0041)**|**0.9228 (0.0041)**  |
>
> | KD on NB301       |   0.5% |  1%    |    5%  |    10%    |    50%   |  100%|
> |----------------------|--------------|--------------|--------------|--------------|--------------|--------------|
> | XGBoost             | 0.2725 (0.0395) | 0.3059 (0.0285) | 0.3313 (0.0120) | 0.3227 (0.0217) | 0.3461 (0.0034) | 0.3766 (0.0000) |
> |TA-GATES         |  **0.5728 (0.0307)**|**0.6351 (0.0138)**  |  **0.7123 (0.0087)**|**0.7331 (0.0071)**|**0.7685 (0.0066)**  | **0.7766 (0.0033)**  |
>
> | NB101 (5%)              |    P@Top 0.1% |P@Top 0.5% |P@Top 1%|P@Top 5%|P@Top 10%|P@Top 50%|P@Top 100%|
> |----------------------|--------------|--------------|--------------|--------------|--------------|--------------|--------------|
> | NASBOWL                 | 0.0476|0.0926|0.1944|0.4341|0.5862|0.8243|1|
> | SemiNAS                | 0.0317|0.2129|0.3132|0.4597|0.5212|0.7995|1 |
> | XGBoost             |0.0476|0.1204|0.1898|0.4515|0.5358|0.8041|1|
> |TA-GATES         |**0.1269**|**0.3518**|**0.4891**|**0.5192**|**0.6201**|**0.8157**|**1**|
>
> | NB201 (5%)              |    P@Top 0.1% |P@Top 0.5% |P@Top 1%|P@Top 5%|P@Top 10%|P@Top 50%|P@Top 100%|
> |----------------------|--------------|--------------|--------------|--------------|--------------|--------------|--------------|
> | NASBOWL                 | 0.0000|0.0614|0.1228|0.4156|0.5841|0.8792|1|
> | XGBoost             |0.1905|0.1966|0.2350|0.4077|0.5450|0.8362|1|
> |TA-GATES         |**0.4762**|**0.4330**|**0.5242**|**0.6829**|**0.7990**|**0.9486**|**1**|
>
> | NB301 (5%)              |    P@Top 0.1% |P@Top 0.5% |P@Top 1%|P@Top 5%|P@Top 10%|P@Top 50%|P@Top 100%|
> |----------------------|--------------|--------------|--------------|--------------|--------------|--------------|--------------|
> | XGBoost             |0.0000|0.0157|0.0542|0.3564|0.5346|0.6595|1 |
> |TA-GATES         |**0.0435**|**0.1224**|**0.2213**|**0.5901**|**0.7585**|**0.8525**|**1**|
>
> | Top-1 error (#Param) | CIFAR-10     | ImageNet     |
> |----------------------|--------------|--------------|
> | CATE                 | 2.46% (4.1M) | 25.0% (5.8M) |
> | NASBOWL | 2.61% (3.7M) | - |
> | TA-GATES             | 2.47% (4.0M) | 24.1% (5.6M) |
>
> As we can see, although there are more surrogate/predictor-related studies in the recent two years, if we solely consider the encoder design and keep other factors the same, GATES is still one of the strongest baseline. Surely, there are plenty of valuable training techniques (e.g., semi-supervised / self-supervised training method) proposed in these studies, which are orthogonal to this work's focus, i.e., a more suitable encoder design for neural architectures. We'll add these results and discussions.

---

### Meta-Review · Area_Chair_paNL · 2022-08-29

**Recommendation:** Accept
**Confidence:** Certain

**Metareview:**

The paper proposes a new encoding scheme for neural architecture search.   All reviewers agreed that operations with different trainable parameters need different encodings.   The paper is clearly written and well-motivated.   It will shed great light on future work to consider the trainable parameter for NAS.
As suggested by dXbo and zsxx, the paper still needs polish.  It should improve the competitor methods, especially providing other strong NAS baselines and  NN encoding schemes.

**Award:**

No

---

### Decision · Program_Chairs · 2022-09-14

Accept